# Small steps no more: Global convergence of stochastic gradient bandits for arbitrary learning rates

**Jincheng Mei**[1]    **Bo Dai**[1 3]    **Alekh Agarwal**[2]    **Sharan Vaswani**[5]    **Anant Raj**[6]
**Csaba Szepesvári**[1 4]    **Dale Schuurmans**[1 4]

[1]Google DeepMind   [2]Google Research   [3]Georgia Institute of Technology   [4]University of Alberta
[5]Simon Fraser University   [6]Indian Institute of Science

{jcmei,bodai,alekhagarwal,szepi,schuurmans}@google.com
vaswani.sharan@gmail.com   anantraj@iisc.ac.in

## Abstract

We provide a new understanding of the stochastic gradient bandit algorithm by showing that it converges to a globally optimal policy almost surely using *any* constant learning rate. This result demonstrates that the stochastic gradient algorithm continues to balance exploration and exploitation appropriately even in scenarios where standard smoothness and noise control assumptions break down. The proofs are based on novel findings about action sampling rates and the relationship between cumulative progress and noise, and extend the current understanding of how simple stochastic gradient methods behave in bandit settings.

## 1   Introduction

The stochastic gradient method has been ubiquitous in the field of machine learning for decades [4]. When applied to reinforcement learning (RL), a representative instantiation of stochastic gradient is the well known policy gradient [32] (or REINFORCE [34]) algorithm, where in each iteration an online sample is gathered using the current policy, from which a gradient estimate is obtained to conduct parameter updates. In the simplest setting of a stochastic bandit problem [16], where decisions matter only for one step, the REINFORCE policy gradient method becomes equivalent to the stochastic gradient bandit algorithm [31, Section 2.8]. Compared to other statistical methods, such as the upper confidence bound algorithm (UCB, [14, 3]), and Thompson sampling (TS, [33, 2]), the stochastic gradient bandit algorithm is conceptually simpler and more computationally efficient, as it does not calculate exploration bonuses nor posterior distributions. Moreover, the stochastic gradient method is highly scalable and naturally applicable to large scale neural networks [29, 30].

However, unlike UCB or TS, the stochastic gradient bandit algorithm does not have an equivalently well established and comprehensive theoretical footing. Given its pervasive success and widespread application in RL [30] and fine-tuning for large language models [26, 27]), it remains an important question to understand the success of stochastic gradient based algorithms in bandit-like settings, not only to bridge the gap between theory and practice, but also to identify more effective and robust variants. In this paper, we make a significant contribution to the theoretical understanding of the stochastic gradient bandit algorithm, bringing its justification closer to that of other less scalable but theoretically well established methods. In particular, we establish the surprising result that:

> *For any constant learning rate $\eta > 0$, the stochastic gradient bandit algorithm is guaranteed to converge to the globally optimal policy almost surely.*

38th Conference on Neural Information Processing Systems (NeurIPS 2024).

Since learning rate is the only tuning parameter in the stochastic gradient bandit algorithm, this result offers a remarkable robustness for the method, that it converges to a near optimal policy, irrespective of the value of this hyperparameter! Analysis of this algorithm is challenging because it requires techniques for simultaneously handling non-convex optimization, stochastic approximation, and the exploration-exploitation trade-off. Prior theoretical work on the stochastic gradient algorithm has primarily focused on non-convex optimization and stochastic approximation, but understanding the simultaneous effect on exploration has been largely lacking.

Recently, significant progress has been made in establishing global convergence results for policy gradient (PG) methods. For example, it has been shown that using exact gradients, Softmax PG converges to a globally optimal policy asymptotically as the number of iterations $t$ goes to infinity [1]. Subsequent work has demonstrated that the asymptotic rate of convergence is $O(1/t)$[23], albeit with problem and initialization dependent constants [22, 17]. The rate and constant dependence in the true gradient setting have been improved via several techniques, including entropy regularization [23], normalization [21], and using natural gradient (mirror descent) [1, 6, 15].

Unfortunately, in the online stochastic setting, where the policy gradient has to be estimated using the current policy to collect samples, these accelerated methods all obtain worse asymptotic results than the standard Softmax PG [20], failing to converge to a global optimum without careful design choices [19]. Yet in the same setting, standard Softmax PG has been shown to succeed in its simplest form, provided only that a sufficiently small learning constant rate $\eta \in \Theta(1)$ is used [24].

For stochastic gradient based methods, decaying or sufficiently small learning rates are used by almost all current approaches, motivated by classical convergence analyses from stochastic optimization [28, 12, 39, 38, 9, 37, 36, 24, 8]. Stationary point convergence is guaranteed for learning rates sufficiently small with respect to the smoothness of the objective function, while also decaying to zero at a precise rate if noise in the gradient estimator persists. In addition to appropriate learning rate control, many other techniques have been developed to control the effects of gradient noise, including regularization [38, 9], variance reduction [37], and carefully considering growth conditions [36, 24].

The technical challenges we face in the current study can be understood in the following aspects: **(1)** Using an arbitrarily large constant learning rate for online stochastic gradient optimization immediately renders the smoothness and noise control techniques mentioned above inapplicable. **(2)** With any constant learning rate $\eta > 0$, the question of whether oscillation or convergence will ultimately occur needs to be addressed before even considering whether any convergence is to a global optimum. This additional level of complexity arises because the optimization objective is not necessarily improved monotonically in expectation. Finally, **(3)** The gradient bandit algorithm does not use any exploration bonus, which means that new techniques are required to demonstrate that it adequately balances the exploration-exploitation trade-off.

In this paper, we resolve the above difficulties by uncovering intriguing exploration properties of stochastic gradient when using any constant learning rate. In particular, we establish the following.

 **(i)** In the stochastic online setting, with probability 1, the stochastic gradient bandit algorithm will not keep sampling any single action forever, implying that it will exhibit a minimal form of exploration without any further modification. This asymptotic event (as $t \to \infty$) happens with probability 1 and holds for any constant learning rate $\eta > 0$.
 **(ii)** This result can then be leveraged to show that, as a consequence, given any constant learning rate, the stochastic gradient bandit algorithms will converge to the globally optimal policy as $t \to \infty$, with probability 1. That is, the probability of sub-optimal actions decays to 0, even though some of them are taken infinitely often asymptotically.

## 2   Setting and Background

We consider the stochastic multi-armed bandit problem [16], specified by $K$ actions and a true mean reward vector $r \in \mathbb{R}^K$, where for each action $a \in [K] := \{1, 2, \ldots, K\}$,

$$r(a) = \int_{-R_{\max}}^{R_{\max}} x \cdot P_a(x)\mu(dx), \tag{1}$$

where $R_{\max} > 0$ is the reward range, $\mu$ is a finite measure over $[-R_{\max}, R_{\max}]$, and $P_a(x) \geq 0$ is the probability density function with respect to $\mu$. We use $R_a$ to denote the reward distribution for

---
**Algorithm 1** Gradient bandit algorithm (without baselines)
---

**Input:** initial parameters $\theta_1 \in \mathbb{R}^K$, learning rate $\eta > 0$.
**Output:** policies $\pi_{\theta_t} = \text{softmax}(\theta_t)$.
**while** $t \geq 1$ **do**
    Sample an action $a_t \sim \pi_{\theta_t}(\cdot)$ and observe reward $R_t(a_t) \sim P_{a_t}$.
    **for** all $a \in [K]$ **do**
      **if** $a = a_t$ **then**
        $\theta_{t+1}(a) \leftarrow \theta_t(a) + \eta \cdot (1 - \pi_{\theta_t}(a)) \cdot R_t(a_t)$.
      **else**
        $\theta_{t+1}(a) \leftarrow \theta_t(a) - \eta \cdot \pi_{\theta_t}(a) \cdot R_t(a_t)$.
      **end if**
    **end for**
**end while**

---

action $a$ defined by the density $P_a$ and base measure $\mu$. The goal is to find a policy $\pi_\theta \in [0,1]^K$ to achieve high expected reward,

$$\max_{\theta \in \mathbb{R}^K} \pi_\theta^\top r, \tag{2}$$

where $\pi_\theta$ is parameterized by $\theta \in \mathbb{R}^K$.

**The gradient bandit algorithm.** A natural idea to optimize Eq. (2) is to use stochastic gradient ascent, which is shown in Algorithm 1 and known as the gradient bandit algorithm [31, Section 2.8]. In Algorithm 1, in each iteration $t \geq 1$, the probability of pulling arm $a \in [K]$ is given as

$$\pi_{\theta_t}(a) = [\text{softmax}(\theta_t)](a) := \frac{\exp\{\theta_t(a)\}}{\sum_{a' \in [K]} \exp\{\theta_t(a')\}}, \quad \text{for all } a \in [K], \tag{3}$$

where $\theta_t \in \mathbb{R}^K$ is the parameter vector to be updated. The following proposition shows that Algorithm 1 is an instance of stochastic gradient ascent with an unbiased gradient estimator [31, 24].

**Proposition 1** (Proposition 2.3 of [24]). *Algorithm 1 is equivalent to the following update,*

$$\theta_{t+1} \leftarrow \theta_t + \eta \cdot \frac{d\, \pi_{\theta_t}^\top \hat{r}_t}{d\theta_t} = \theta_t + \eta \cdot \left( diag(\pi_{\theta_t}) - \pi_{\theta_t} \pi_{\theta_t}^\top \right) \hat{r}_t, \tag{4}$$

*where $\mathbb{E}_t \left[ \frac{d\, \pi_{\theta_t}^\top \hat{r}_t}{d\theta_t} \right] = \frac{d\, \pi_{\theta_t}^\top r}{d\theta_t}$, and $\mathbb{E}_t[\cdot]$ is defined with respect to randomness from on-policy sampling $a_t \sim \pi_{\theta_t}(\cdot)$ and reward sampling $R_t(a_t) \sim P_{a_t}$. The Jacobian of $\theta \mapsto \pi_\theta := \text{softmax}(\theta)$ is $\left( \frac{d\, \pi_\theta}{d\theta} \right)^\top = diag(\pi_\theta) - \pi_\theta \pi_\theta^\top \in \mathbb{R}^{K \times K}$, and $\hat{r}_t(a) := \frac{\mathbb{I}\{a_t = a\}}{\pi_{\theta_t}(a)} \cdot R_t(a)$ for all $a \in [K]$ is the importance sampling (IS) estimator, and we set $R_t(a) = 0$ for all $a \neq a_t$.*

**Known results on the convergence of the gradient bandit algorithm.** Since Eq. (2) corresponds to a smooth non-concave maximization problem over $\theta \in \mathbb{R}^K$ [23], using Algorithm 1 with decaying learning rates is sufficient to guarantee convergence to a stationary point [28, 25, 12, 39]. However, this is insufficient to ensure the globally optimal solution of Eq. (2) is reached, since there exist multiple stationary points. More recently, guarantees of convergence to a globally optimal policy have been developed for PG methods in the true gradient setting [1, 23], where the algorithm has access to exact mean rewards. These results were later extended to achieve global convergence guarantees (almost surely) in the stochastic setting [38, 9, 37, 36, 24, 18]. However, these extended results have required decaying or sufficiently small learning rates, motivated by exploiting smoothness and combating the inherent noise in stochastic gradients.

Despite these previous assumptions, there exists empirical and theoretical evidence that using a large learning rate in the stochastic gradient bandit algorithm is a viable option. For example, it has been observed that softmax policies learn even with extremely large learning rates such as $2^{14}$ [11]. For logistic regression on linearly separable data, the objective has an exponential tail and the minimizer is unbounded, yet it has been shown that gradient descent with iteration dependent learning rate $\eta \in \Theta(t)$ achieves accelerated $O(1/t^2)$ convergence [35]. Though the objective in Eq. (2) has

similar properties, unlike logistic regression, the problem we are considering is non-concave, so the same techniques cannot be directly applied. The most related results are from [24], which proved that with a small problem specific constant learning rate, Algorithm 1 achieves convergence to a globally optimal policy almost surely. However, as mentioned, the learning rate choices in [24] rely on assumptions of (non-uniform) smoothness and noise growth conditions (their Lemmas 4.2, 4.3, and 4.6), which cannot be directly applied here for a large learning rate.

Consequently, the use of large learning rates appear to render existing results and techniques inapplicable. Furthermore, with a large constant learning rate, it is unclear whether Algorithm 1 will converge to any stationary point, or the iterates will keep oscillating. If the algorithm does converge, it is also not clear what effect large step-sizes have on exploration, and whether the algorithm will converge to the optimal arm in such cases. Resolving these questions requires new results that characterize the behavior of Algorithm 1, since the classical optimization and stochastic approximation convergence theories are no longer applicable, as explained.

# 3 Asymptotic Global Convergence of Gradient Bandit Algorithm

We have seen that solving the non-concave maximization problem Eq. (2) using Algorithm 1 with any constant (potentially large) learning rate requires ideas beyond classical optimization theory. Here, we take a different perspective to investigate how Algorithm 1 samples actions. For analysis, we make the following assumption about the reward distribution.

**Assumption 1** (True mean reward has no ties). *For all $i, j \in [K]$, if $i \neq j$, then $r(i) \neq r(j)$.*

**Remark 1.** *Removing Assumption 1 remains an open question for future work, while we believe that Algorithm 1 works without Assumption 1. One piece of evidence to support this conjecture is that even in the exact gradient setting, the set of initializations where Softmax PG approaches non-strict one-hot policies has zero measure.*

## 3.1 Failure Mode of Aggressive Updates

It has been observed that several accelerated PG methods in the true gradient setting, including natural PG [13, 1] and normalized PG [21], obtain worse results than standard softmax PG if combined with online sampling $a_t \sim \pi_{\theta_t}(\cdot)$ using constant learning rates [20]. The failure mode in these cases is that the update is too aggressive and commits to a sub-optimal arm without sufficiently exploring all arms. This results in a non-trivial probability of sampling one action forever, i.e., there exists a potentially sub-optimal action $a \in [K]$, such that with some constant probability, $a_t = a$ for all $t \geq 1$. Such an outcome implies that $\pi_{\theta_t}(a) \to 1$ as $t \to \infty$ [20, Theorem 3]. Since $a \in [K]$ could be a sub-optimal action with $r(a) < r(a^*) = \max_{a \in [K]} r(a)$, this results in a lack of exploration, and consequently, methods such as natural PG and normalized PG are not guaranteed to converge to the optimal action $a^* := \arg\max_{a \in [K]} r(a)$ with probability 1.

## 3.2 Stochastic Gradient Automatically Avoids Lack of Exploration

Our first key finding is that Algorithm 1 does not keep sampling one action forever, no matter how large the constant learning rate is. This property avoids the problem of a lack of exploration, in the sense that Algorithm 1 will at least explore more than one action infinitely often. At first glance, this might not seem like a strong property, since the algorithm might somehow explore only sub-optimal actions forever. However, we will argue below that this property coupled with additional arguments is sufficient to guarantee convergence to the globally optimal policy.

Let us now formally prove the above property. By Algorithm 1, for all $a \in [K]$, for all $t \geq 1$,

$$\theta_{t+1}(a) \leftarrow \theta_t(a) + \begin{cases} \eta \cdot (1 - \pi_{\theta_t}(a)) \cdot R_t(a), & \text{if } a_t = a, & (5) \\ -\eta \cdot \pi_{\theta_t}(a) \cdot R_t(a_t), & \text{otherwise.} & (6) \end{cases}$$

We define $N_t(a)$ as the number of times action $a \in [K]$ is sampled up to iteration $t \geq 1$, i.e.,

$$N_t(a) := \sum_{s=1}^{t} \mathbb{I}\{a_s = a\}, \tag{7}$$

and its asymptotic limit $N_\infty(a) := \lim_{t\to\infty} N_t(a)$, which could possibly be infinity. For all $a \in [K]$, we have either $N_\infty(a) = \infty$ or $N_\infty(a) < \infty$, meaning that $a \in [K]$ is sampled infinitely often or only finitely many times asymptotically. First, we prove the following Lemma 1, which shows that if an action $a \in [K]$ is sampled only finitely many times as $t \to \infty$, then the parameter corresponding to action $a$ is also finite, i.e., $\sup_{t\geq 1} |\theta_t(a)| < \infty$.

**Lemma 1.** *Using Algorithm 1 with any constant $\eta \in \Theta(1)$, if $N_\infty(a) < \infty$ for an action $a \in [K]$, then we have, almost surely,*

$$\sup_{t\geq 1} \theta_t(a) < \infty, \text{ and } \inf_{t\geq 1} \theta_t(a) > -\infty. \tag{8}$$

Lemma 1 will be used multiple times in the subsequent convergence arguments.

**Proof sketch.** Since we assume action $a \in [K]$ is sampled finitely many times, the update given in the case depicted by Eq. (5) happens finitely many times. Each update is bounded since the sampled reward is in $[-R_{\max}, R_{\max}]$ by Eq. (1), and the learning rate is a constant, i.e., $\eta \in \Theta(1)$. In Algorithm 1, $\theta_t(a)$ is still updated even when $a_t \neq a$, with the corresponding update given by the case depicted by Eq. (6). Therefore, whether $\theta_t(a)$ is bounded depends on the cumulative probability $\sum_{s=1}^{t} \pi_{\theta_s}(a)$ being summable as $t \to \infty$. According to the extended Borel-Cantelli lemma (Lemma 3), we have, almost surely,

$$\left\{ \sum_{t\geq 1} \pi_{\theta_t}(a) = \infty \right\} = \{ N_\infty(a) = \infty \}, \tag{9}$$

which implies (by taking complements) that $\sum_{t\geq 1} \pi_{\theta_t}(a) < \infty$ if and only if $N_\infty(a) < \infty$. Therefore, if $a \in [K]$ is sampled finitely often, $\theta_t(a)$ will be updated in a bounded manner (using Eqs. (5) and (6)) as $t \to \infty$, hence establishing Lemma 1. Detailed proofs for this lemma, as well as for all other results in this paper can be found in the appendix.

Given Lemma 1, we can then establish the above-mentioned finding about the exploration effect of Algorithm 1 in Lemma 2.

**Lemma 2** (Avoiding a lack of exploration). *Using Algorithm 1 with any $\eta \in \Theta(1)$, there exists at least a pair of distinct actions $i, j \in [K]$ and $i \neq j$, such that, almost surely,*

$$N_\infty(i) = \infty, \text{ and } N_\infty(j) = \infty. \tag{10}$$

**Proof sketch.** The argument for the existence of one such action is straightforward, since by the pigeonhole principle, if there are finitely many actions, i.e., $K < \infty$, there must be at least one action $i \in [K]$ that is sampled infinitely often as $t \to \infty$.

The argument for the existence of a second such action is by contradiction. Suppose that all the other actions $j \in [K]$ with $j \neq i$ are sampled only finitely many times as $t \to \infty$. According to Lemma 1, their corresponding parameters must remain finite, i.e., $\sup_{t\geq 1} |\theta_t(j)| < \infty$ for all $j \in [K]$ with $j \neq i$. Now consider $\theta_t(i)$. By assumption, the second update case for this parameter, Eq. (6), happens only finitely often, since Eq. (6) can only occur when $a_t \neq i$. Therefore, the key question is whether the cumulative probability $\sum_{s=1}^{t} (1 - \pi_{\theta_s}(i))$ involved in the first case of the update, Eq. (5), is summable as $t \to \infty$. Note that $\sum_{s=1}^{t} (1 - \pi_{\theta_s}(i)) = \sum_{s=1}^{t} \sum_{j\neq i} \pi_{\theta_s}(j)$, which is indeed summable as $t \to \infty$, by the assumption and Eq. (9). This implies that action $i$, which is sampled infinitely often, achieves a parameter magnitude, $\sup_{t\geq 1} |\theta_t(i)| < \infty$, that remains bounded as $t \to \infty$. Using the softmax parameterization Eq. (3) in the above argument, we conclude that for all $a \in [K]$, $\inf_{t\geq 1} \pi_{\theta_t}(a) > 0$, i.e., every action's probability remains bounded away from zero, and hence is not summable. Using Eq. (9), this implies that every action is sampled infinitely often, which contradicts the assumption that only action $i$ is sampled infinitely often as $t \to \infty$.

**Discussion.** Lemma 2 implies that Algorithm 1 is not an aggressive method in the sense of [20], no matter how large the learning rate is, as long as it is constant, i.e., $\eta \in \Theta(1)$. According to [20, Theorem 7], even if we fix the sampling in Algorithm 1 to a sub-optimal action $a \in [K]$ forever, i.e., $a_t = a$ for all $t \geq 1$, its probability will not approach 1 faster than $O(1/t)$, i.e., $1 - \pi_{\theta_t}(a) \in \Omega(1/t)$. This means that there must be at least one another action $a' \in [K]$ with $a' \neq a$, such that $a'$ will also be sampled infinitely often. A more intuitive explanation is that the $(1 - \pi_{\theta_t}(a))$ term in Eq. (5) will

be near 0, which slows the speed of committing to a deterministic policy on $a$ whenever $\pi_{\theta_t}(a)$ is close to 1, which encourages exploration. Such natural exploratory behavior arises in Algorithm 1 because of the softmax Jacobian $\mathrm{diag}(\pi_\theta) - \pi_\theta \pi_\theta^\top$ in the update shown in Proposition 1, which determines the growth order of $\theta_t(a)$ for all $a \in [K]$ as $t \to \infty$, making the effect of a constant learning rate $\eta \in \Theta(1)$ asymptotically inconsequential.

## 3.3 Warm up: Global Asymptotic Convergence when $K = 2$

We now consider the simplest case, where we have only two possible actions. According to Lemma 2, each of the two actions must be sampled infinitely often as $t \to \infty$. We now illustrate the second key result, that for both actions $a \in [K]$, the random sequence $\{\theta_t(a)\}_{t \geq 1}$ follows the direction of the expected gradient for sufficiently large $t \geq 1$ almost surely. The proof uses a technique that has been previously used in [19, 24] for small learning rates, but here we observe that the same technique continues to work for Algorithm 1 no matter how large the learning rate is, as long as $\eta \in \Theta(1)$.

**Theorem 1.** *Let $K = 2$ and $r(1) > r(2)$. Using Algorithm 1 with any $\eta \in \Theta(1)$, we have, almost surely, $\pi_{\theta_t}(a^*) \to 1$ as $t \to \infty$, where $a^* := \arg\max_{a \in [K]} r(a)$ (equal to Action 1 in this case).*

**Proof sketch.** According to Lemma 2, $N_\infty(1) = N_\infty(2) = \infty$. Denote the the reward gap as $\Delta := r(a^*) - \max_{a \neq a^*} r(a) > 0$, which becomes $\Delta = r(1) - r(2)$ for two actions. Since the stochastic gradient is unbiased (Proposition 1), we have, for all $t \geq 1$ (detailed calculations omitted),

$$\mathbb{E}_t[\theta_{t+1}(a^*)] = \theta_t(a^*) + \eta \cdot \pi_{\theta_t}(a^*) \cdot \left(r(a^*) - \pi_{\theta_t}^\top r\right) \tag{11}$$

$$= \theta_t(a^*) + \eta \cdot \pi_{\theta_t}(a^*) \cdot \Delta \cdot (1 - \pi_{\theta_t}(a^*)) > \theta_t(a^*). \tag{12}$$

A similar calculation shows that,

$$\mathbb{E}_t[\theta_{t+1}(2)] = \theta_t(2) - \eta \cdot \pi_{\theta_t}(2) \cdot \Delta \cdot (1 - \pi_{\theta_t}(2)) < \theta_t(2), \tag{13}$$

which means that $\theta_t(a^*)$ is monotonically increasing in expectation and $\theta_t(2)$ is monotonically decreasing in expectation. In other words, $\{\theta_t(a^*)\}_{t \geq 1}$ is a sub-martingale, while $\{\theta_t(2)\}_{t \geq 1}$ is a super-martingale. However, since $\theta_t \in \mathbb{R}^K$ is unbounded, Doob's martingale convergence results cannot be directly applied, so we pursue a different argument. Following [19, 24], given an action $a \in [K]$, we define $P_t(a) := \mathbb{E}_t[\theta_{t+1}(a)] - \theta_t(a)$ as the "progress", and define $W_t(a) := \theta_t(a) - \mathbb{E}_{t-1}[\theta_t(a)]$ as the "noise", where $\theta_t(a) = W_t(a) + P_{t-1}(a) + \theta_{t-1}(a)$. By recursion we can determine that,

$$\theta_t(a) = \mathbb{E}[\theta_1(a)] + \sum_{s=1}^{t} W_s(a) + \sum_{s=1}^{t-1} P_s(a), \tag{14}$$

i.e., $\theta_t(a)$ is the result of "cumulative progress" and "cumulative noise". According to [24, Theorem C.3], the cumulative noise term can be bounded by using martingale concentration, where the order of the corresponding confidence interval is smaller than the order of the cumulative progress. Therefore, the summation will always be determined by the cumulative progress as $t \to \infty$. According to the calculations in Eqs. (12) and (13), we have $P_t(a^*) > 0$ and $P_t(2) < 0$, both of which are not summable. As a result, $\theta_t(a^*) \to \infty$ and $\theta_t(2) \to -\infty$ as $t \to \infty$, which implies that $\frac{\pi_{\theta_t}(a^*)}{\pi_{\theta_t}(2)} = \exp\{\theta_t(a^*) - \theta_t(a)\} \to \infty$, hence $\pi_{\theta_t}(a^*) \to 1$ as $t \to \infty$.

## 3.4 Global Asymptotic Convergence for all $K \geq 2$

The illustrative two-action case shows that if $\pi_{\theta_t}^\top r \in (r(2), r(a^*))$ and if both actions are sampled infinitely often, then we have, almost surely $\theta_t(a^*) \to \infty$ and $\theta_t(2) \to -\infty$ as $t \to \infty$. However, the question at the beginning of Section 3.2 remains: when $K > 2$, if the two actions sampled infinitely often in Lemma 2 are both sub-optimal, will that result in a similar failure mode to the one described in Section 3.1? The answer is no, which follows from our third key finding, which is based on another contradiction-based argument that establishes almost sure convergence to a globally optimal policy in the general $K > 2$ case.

**Theorem 2.** *Given $K \geq 2$, using Algorithm 1 with any $\eta \in \Theta(1)$, we have, almost surely, $\pi_{\theta_t}(a^*) \to 1$ as $t \to \infty$, where $a^* = \arg\max_{a \in [K]} r(a)$ is the optimal action.*

**Proof sketch.** We consider two cases: $N_\infty(a^*) < \infty$ and $N_\infty(a^*) = \infty$, corresponding to whether the optimal action is sampled finitely or infinitely often as $t \to \infty$. We argue that the first case ($N_\infty(a^*) < \infty$) is impossible, while for the second case ($N_\infty(a^*) = \infty$) we prove that $\theta_t(a^*) - \theta_t(a) \to \infty$ for all $a \in [K]$ with $r(a) < r(a^*)$, which implies $\pi_{\theta_t}(a^*) \to 1$ as $t \to \infty$.

*First case.* Suppose that $N_\infty(a^*) < \infty$. We argue that this is impossible via contradiction. Given the assumption and Lemma 2 we know there must be at least two other sub-optimal actions $i_1, i_2 \in [K]$, $i_1 \neq i_2$, such that $N_\infty(i_1) = N_\infty(i_2) = \infty$. In particular, let $i_1 = \arg\min_{a \in [K], N_\infty(a)=\infty} r(a)$ and $i_2 = \arg\max_{a \in [K], N_\infty(a)=\infty} r(a)$, hence $r(i_1) < r(i_2) < r(a^*)$. By Lemma 6 (see Appendix) we will also have $r(i_1) < \pi_{\theta_t}^\top r < r(i_2)$ for sufficiently large $t \geq 1$, which implies for action $i_1$,

$$\mathbb{E}_t[\theta_{t+1}(i_1)] = \theta_t(i_1) + \eta \cdot \pi_{\theta_t}(i_1) \cdot \big(r(i_1) - \pi_{\theta_t}^\top r\big) < \theta_t(i_1), \qquad (15)$$

for sufficiently large $t \geq 1$, which further implies that $\sup_{t \geq 1} \theta_t(i_1) < \infty$. Meanwhile, for the optimal action $a^*$, the assumption and Lemma 1 imply that $\inf_{t \geq 1} \theta_t(a^*) > -\infty$. Combining these two observations gives,

$$\sup_{t \geq 1} \frac{\pi_{\theta_t}(i_1)}{\pi_{\theta_t}(a^*)} = \sup_{t \geq 1} \exp\{\theta_t(i_1) - \theta_t(a^*)\} < \infty. \qquad (16)$$

On the other hand, since $N_\infty(i_1) = \infty$ and $N_\infty(a^*) < \infty$ (by assumption), we then have $\sup_{t \geq 1} \exp\{\theta_t(i_1) - \theta_t(a^*)\} = \infty$ by Lemma 5 (see Appendix), which contradicts Eq. (16).

*Second case.* Suppose that $N_\infty(a^*) = \infty$. We will argue that $\pi_{\theta_t}(a^*) \to 1$ as $t \to \infty$ almost surely. First, according to Lemma 2, there exists at least one sub-optimal action $i_1 \in [K]$, $i_1 \neq a^*$, such that $N_\infty(i_1) = \infty$. Let $i_1 = \arg\min_{a \in [K], N_\infty(a)=\infty} r(a)$. By Lemma 6 and the definition of $a^*$, we have $r(i_1) < \pi_{\theta_t}^\top r < r(a^*)$ for all sufficiently large $t \geq 1$. Since $N_\infty(i_1) = \infty$, using similar calculations to Eqs. (13) and (14) in Theorem 1, we have, $\theta_t(i_1) \to -\infty$ as $t \to \infty$. We also have, $\inf_{t \geq 1} \theta_t(a^*) > -\infty$ as $t \to \infty$. Hence, $\frac{\pi_{\theta_t}(a^*)}{\pi_{\theta_t}(i_1)} = \exp\{\theta_t(a^*) - \theta_t(i_1)\} \to \infty$ as $t \to \infty$.

Define $\mathcal{A}_\infty := \{a \in [K] \mid N_\infty(a) = \infty\}$ as the set of actions that are sampled infinitely often, and note that $|\mathcal{A}_\infty| \geq 2$ by Lemma 2. Sort the action indices in $\mathcal{A}_\infty$ according to their expected reward values in descending order, i.e.,

$$r(a^*) > r(i_{|\mathcal{A}_\infty|-1}) > r(i_{|\mathcal{A}_\infty|-2}) > \cdots > r(i_2) > r(i_1). \qquad (17)$$

Assumption 1 is used here to prevent two arms from having the same reward and thus guarantee the inequalities are strict in Eq. (17). Next, using similar calculations as in Lemma 6, we have,

$$\pi_{\theta_t}^\top r - r(i_2) > \pi_{\theta_t}(a^*) \cdot \left[ r(a^*) - r(i_2) - \sum_{a^- \in \mathcal{A}^-(i_2)} \frac{\pi_{\theta_t}(a^-)}{\pi_{\theta_t}(a^*)} \cdot (r(i_2) - r(a^-)) \right], \qquad (18)$$

where $\mathcal{A}^-(i_2) := \{a^- \in [K] : r(a^-) < r(i_2)\}$ is the set of actions that have lower mean reward than $i_2 \in [K]$, and note that $i_1 \in \mathcal{A}^-(i_2)$. Using the above definitions, we can conclude that $i_1$ is the only arm in $\mathcal{A}^-(i_2)$ that has been sampled infinitely often. According to Lemma 5, for all $a^- \in \mathcal{A}^-(i_2)$ with $a^- \neq i_1$, we have, $\frac{\pi_{\theta_t}(a^*)}{\pi_{\theta_t}(a^-)} \to \infty$ as $t \to \infty$, since $N_\infty(a^*) = \infty$ (by assumption) and $N_\infty(a^-) < \infty$ (by Eq. (17)). Therefore, for all sufficiently large $t$, the probability ratio in Eq. (18) $\frac{\pi_{\theta_t}(a^*)}{\pi_{\theta_t}(a^-)} \to \infty$ for all $a^- \in \mathcal{A}^-(i_2)$, which implies that, for all sufficiently large $t \geq 1$,

$$\pi_{\theta_t}^\top r - r(i_2) > 0.5 \cdot \pi_{\theta_t}(a^*) \cdot \big(r(a^*) - r(i_2)\big) > 0. \qquad (19)$$

We have thus shown that $\pi_{\theta_t}^\top r > r(i_2)$. Recall that we had previously proved that $\pi_{\theta_t}^\top r > r(i_1)$. Hence, we will apply this argument recursively: after this point, $i_2 \in [K]$ will become the new "$i_1 \in [K]$", and a similar inequality to Eq. (13) will then hold for $i_2 \in [K]$ from similar calculations to Eqs. (13) and (14) in Theorem 1, establishing $\theta_t(i_2) \to -\infty$ as $t \to \infty$. This will imply that for all sufficiently large $t \geq 1$,

$$\pi_{\theta_t}^\top r - r(i_3) > 0.5 \cdot \pi_{\theta_t}(a^*) \cdot \big(r(a^*) - r(i_3)\big) > 0. \qquad (20)$$

Continuing the recursive argument, we can conclude for all actions $a \in \mathcal{A}_\infty$ with $a \neq a^*$ that $\frac{\pi_{\theta_t}(a^*)}{\pi_{\theta_t}(a)} \to \infty$ as $t \to \infty$. Meanwhile, for all actions $a \notin \mathcal{A}_\infty$, Lemma 5 also shows that $\frac{\pi_{\theta_t}(a^*)}{\pi_{\theta_t}(a)} \to \infty$ as $t \to \infty$. Combining these two results yields the conclusion that for all sub-optimal actions $a \in [K]$ with $r(a) < r(a^*)$ we have $\frac{\pi_{\theta_t}(a^*)}{\pi_{\theta_t}(a)} \to \infty$ as $t \to \infty$, which implies $\pi_{\theta_t}(a^*) \to 1$ as $t \to \infty$. Thus, we have established almost sure convergence to the globally optimal policy.

**Discussion.** Lemma 2 is important to prove that the optimal arm will be sampled infinitely often. In particular, Lemma 2 guarantees $|\mathcal{A}_\infty| \geq 2$ and the existence of $i_1$ in Eq. (15), which can then be used to construct the contradiction in Eq. (16). Without Lemma 2, $|\mathcal{A}_\infty|$ might be equal to 1 and the failure mode in Section 3.1 can occur, resulting in Algorithm 1 not sampling the optimal action infinitely often as $t \to \infty$.

### 3.5 Asymptotic Rate of Convergence

According to Theorem 2, almost surely, $\pi_{\theta_t}(a^*) \to 1$ as $t \to \infty$. Therefore, after a large enough time $\tau < \infty$, we have $\pi_{\theta_t}(a^*) \geq 1/2$, which implies that the "progress" term in Eq. (14) can be lower bounded. With this, an asymptotic rate of convergence can be proved as follows.

**Theorem 3.** *For a large enough $\tau > 0$, for all $T > \tau$, the average sub-optimality decreases at an* $O\left(\frac{\ln(T)}{T}\right)$ *rate. Formally, if $a^*$ is the optimal arm, then, for a constant $c$,*

$$\frac{\sum_{s=\tau}^{T} r(a^*) - \langle \pi_s, r \rangle}{T} \leq \frac{c \ln(T)}{T - \tau}.$$

## 4 Simulation Study

To validate and enhance the theoretical findings, we ran experiments with a four action bandit environment ($K = 4$) with a true mean reward vector of $r = (0.2, 0.05, -0.1, -0.4)^\top \in \mathbb{R}^4$. The reward distribution $P_a$ for arm $1 \leq a \leq 4$ is Gaussian, centered at $r(a)$ and with a standard deviation of $0.1$. The environment is chosen to illustrate various phenomenon, which we discuss after presenting the results. The algorithm is Algorithm 1 with $\theta_1 = \mathbf{0} \in \mathbb{R}^K$.

For comparison, assuming that the random rewards belong to the $[-1, 1]$ interval, the only result for the stochastic gradient bandit algorithm [24, Lemma 4.6] that allowed a constant learning rate required that the learning rate be less than $\eta_c = \frac{\Delta^2}{40 \cdot K^{3/2} \cdot R_{\max}^3} = \frac{9}{128000} \approx 0.00007$, where we used $\Delta = 0.15$, $K = 4$, and $R_{\max} = 1$. While technically, the result does not apply to our case where the reward distributions have unbounded support, the probability of the reward landing outside of $[-1, 1]$ is in the order of $10^{-9}$. Choosing $R_{\max}$ to be larger, this probability falls extremely quickly, which suggests that the above threshold is generous. For the experiments we use the learning rates $\eta \in \{1, 10, 100, 1000\}$, that are several orders of magnitudes larger than $\eta_c$. For each learning rate, we plot the outcome of 10 runs, corresponding to different random seeds. Each run lasts $10^6 (\approx e^{14})$ iterations. The log-suboptimality gaps for the $4 \times 10$ cases are shown on Figures 1a-1d, where they are plotted against the logarithm of time. Additional results for $K = 2$ arms are shown in Appendix D. Note that in this example small sub-optimality implies that the optimal arm is chosen with high probability. In what follows, we discuss the results in the plots.

**Asymptotic convergence.** For the smaller learning rates of $\eta = 1$ and $10$, all 10 seeds rapidly and steadily converge, reaching a sub-optimality of $e^{-14}$ or less. For $\eta = 100$ and $1000$, most of the runs reach even small error even faster, but some runs are "stuck" even after $10^6$ steps. Note that this does not contradict the theoretical result; nor do we suspect numerical issues. As seen for the case of $\eta = 100$, even after a long phase with little to no progress, a run can "recover" (see the grey curve). In fact, it is reasonable to expect that the price of increasing the learning rate is larger variance; as seen in these plots (subplots (a) and (b) are also attesting to this). Differences between learning rates are further discussed below.

**Non-monotone objective value.** Using a very small learning rate guarantees monotonic improvement (in expectation) in the policy's expected reward [24]. Conversely, a large learning rate results in non-monotonic evolution of the expected rewards $\{\pi_{\theta_t}^\top r\}_{t \geq 1}$, even in the final stages of convergence, as can be seen clearly for the learning rates of $\eta = 1$ and $10$ in Figures 1a and 1b. For larger $\eta$, the non-monotone behavior happens over longer periods and is less visible in the plots. This is because using large learning rates causes the policy to rapidly increase the parameters for some action, after which the gradient becomes small, limiting further progress.

**Rate of convergence.** Figures 1a and 1b, where the log-log plot has a slope of nearly $-1$, give some evidence that an $O(1/t)$ asymptotic rate is achieved. In general, such a rate cannot be improved

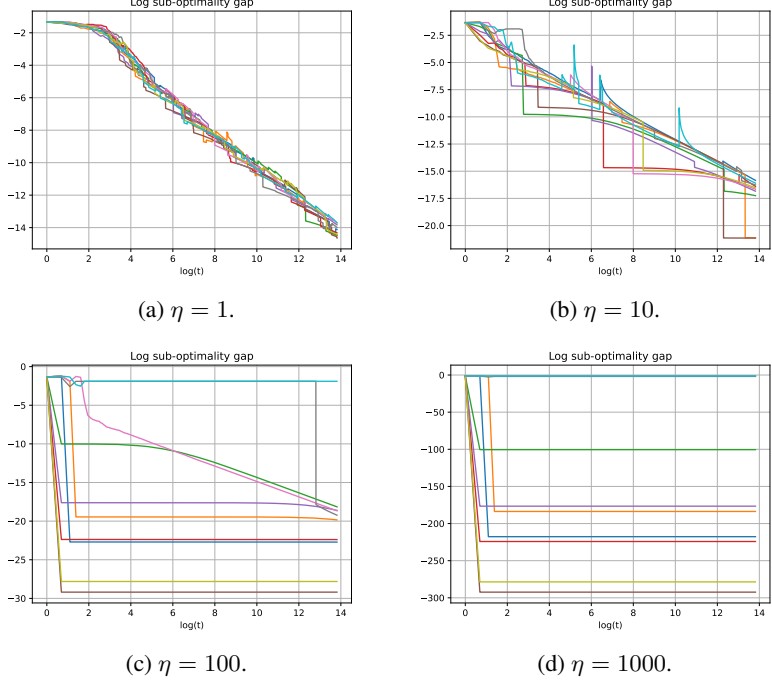



(a) $\eta = 1$.        (b) $\eta = 10$.

(c) $\eta = 100$.        (d) $\eta = 1000$.



Figure 1: Log sub-optimality gap, $\log\left(r(a^*) - \pi_{\theta_t}^\top r\right)$, plotted against the logarithm of time, $\log t$, in a $4$-action problem with various learning rates, $\eta$. Each subplot shows a run with a specific learning rate. The curves in a subplot correspond to 10 different random seeds. Theory predicts that essentially all seeds will lead to a curve converging to zero ($-\infty$ in these plots). For a discussion of the results, see the text.,

in terms of $t$ [14]. Theorem 3 gives a weaker version of convergence rate over averaged iterates (not last iterate), which is slightly worse than $O(1/t)$. More work is needed to verify if the asymptotic convergence rate in Theorem 3 is improvable or not.

**Different learning rates.** Two observations can be made from Figure 1 regarding the effect of using different $\eta$ values: **First**, during the final stage of convergence when $r(a^*) - \pi_{\theta_t}^\top r \approx 0$, using larger $\eta$ results in faster convergence on average. As $\eta$ increases, the order of $\log\left(r(a^*) - \pi_{\theta_t}^\top r\right)$ also changes from $e^{-14}$ ($\eta = 1$), to $e^{-20}$ ($\eta = 100$), and $e^{-200}$ ($\eta = 1000$). We conjecture that the asymptotic rate of convergence has an $O(1/\eta)$ dependence. **Second**, using larger learning rates can take a longer time to enter the final stage of convergence. When $\eta = 1$ or $10$, all curves quickly enter the final stage of $r(a^*) - \pi_{\theta_t}^\top r \approx 0$. However, for larger $\eta$ values, $1/10$ runs ($\eta = 100$) and $3/10$ runs ($\eta = 1000$) result in $r(a^*) - \pi_{\theta_t}^\top r$ values far from $0$ even after $10^6$ iterations. These runs take orders of magnitude more iterations to eventually achieve $r(a^*) - \pi_{\theta_t}^\top r \approx 0$. These situations correspond to the policy $\pi_{\theta_t}$ getting stuck near sub-optimal corners of the simplex, meaning that $\pi_{\theta_t}(i) \approx 1$ for a sub-optimal action $i \in [K]$ with $r(i) < r(a^*)$. In such cases, even Softmax PG with the true gradient can remain stuck on a sub-optimal plateau for an extremely long time [23]. However, the reason why larger learning rates lead to longer plateaus in the stochastic setting remains unclear.

**Trade-offs and multi-stage chracterizations of convergence.** Given the above observations, there appears to exist a trade-off for $\eta$: larger $\eta$ values result in faster convergence during the final stage where $r(a^*) - \pi_{\theta_t}^\top r \approx 0$, but at the the cost of taking far longer to enter this final stage of convergence. Since asymptotic convergence results are insufficient for explaining these subtleties in a satisfactory manner, a more refined analysis that considers the different stages of convergence is required.

# 5    Conclusions and Future Directions

This work refines our understanding of stochastic gradient bandit algorithms by proving that it converges to a globally optimal policy almost surely with *any* constant learning rate. Our new proof strategy based on the asymptotics of sample counts opens new directions for better characterizing exploration effects of stochastic gradient methods, while also suggesting interesting new questions. Characterizing the multiple stages of convergence remains another interesting future direction. One interesting possibility is that there might exist an optimal time-dependent scheme for *increasing* the learning rate (such as $\eta \in O(\log t)$) to accelerate convergence, rather than use a constant $\eta \in O(1)$. This is corroborated by our experiments: As seen in Figure 1, small learning rates perform better during the early stages of optimization, while larger learning rates achieve faster convergence during the final stage. Other directions include extending our bandit results to the more general RL setting [34], as well as extending our results for the softmax tabular parameterization to handle function approximation [1].

**Limitations:** While this work establishes a surprising asymptotic convergence result for any constant learning rate, it does not shed light on the effect of different learning rates on the convergence. Moreover, our analysis is limited to multi-armed bandits, and does not immediately extend to the general RL setting. These aspects are the main limitations of this paper.

**Broader impact:** This is primarily theoretical work on a fundamental algorithm that is used broadly in RL applications. We expect these results to improve the research community's understanding of the basic stochastic gradient bandit method.

## Acknowledgments and Disclosure of Funding

Jincheng Mei would like to thank Ramki Gummadi for providing feedback on a draft of this manuscript. Csaba Szepesvári and Dale Schuurmans gratefully acknowledge funding from the Canada CIFAR AI Chairs Program, Amii and NSERC. Sharan Vaswani acknowledges the support from the NSERC Discovery Grant (RGPIN-2022-04816).

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

# A    Asymptotic Convergence

**Lemma 1.** Using Algorithm 1 with any constant $\eta \in \Theta(1)$, if $N_\infty(a) < \infty$ for an action $a \in [K]$, then we have, almost surely,

$$\sup_{t \geq 1} \theta_t(a) < \infty, \text{ and } \inf_{t \geq 1} \theta_t(a) > -\infty. \tag{21}$$

*Proof.* Suppose $N_\infty(a) < \infty$ for an action $a \in [K]$. According to Algorithm 1,

$$\theta_{t+1}(a) \leftarrow \theta_t(a) + \begin{cases} \eta \cdot (1 - \pi_{\theta_t}(a)) \cdot R_t(a), & \text{if } a_t = a\,, \\ -\eta \cdot \pi_{\theta_t}(a) \cdot R_t(a_t), & \text{otherwise}\,, \end{cases} \tag{22}$$

and let

$$I_t(a) := \begin{cases} 1, & \text{if } a_t = a\,, \\ 0, & \text{otherwise}\,. \end{cases} \tag{23}$$

According to Eq. (22), we have, for all $t \geq 1$,

$$\theta_t(a) - \theta_1(a) = \sum_{s=1}^{t-1} I_s(a) \cdot \eta \cdot (1 - \pi_{\theta_s}(a)) \cdot R_s(a) + \sum_{s=1}^{t-1} (1 - I_s(a)) \cdot (-\eta) \cdot \pi_{\theta_s}(a) \cdot R_s(a_s). \tag{24}$$

Using triangle inequality, we have,

$$|\theta_t(a) - \theta_1(a)| \leq \sum_{s=1}^{t-1} \left| I_s(a) \cdot \eta \cdot (1 - \pi_{\theta_s}(a)) \cdot R_s(a) \right| + \sum_{s=1}^{t-1} \left| (1 - I_s(a)) \cdot (-\eta) \cdot \pi_{\theta_s}(a) \cdot R_s(a_s) \right| \tag{25}$$

$$\leq \eta \cdot R_{\max} \cdot \sum_{s=1}^{t-1} I_s(a) + \eta \cdot R_{\max} \cdot \sum_{s=1}^{t-1} \pi_{\theta_s}(a) \tag{26}$$

$$= \eta \cdot R_{\max} \cdot \left( N_{t-1}(a) + \sum_{s=1}^{t-1} \pi_{\theta_s}(a) \right). \tag{27}$$

By assumption, we have,

$$N_\infty(a) := \lim_{t \to \infty} N_t(a) < \infty. \tag{28}$$

According to the extended Borel-Cantelli Lemma 3, we have, almost surely,

$$\sum_{t=1}^{\infty} \pi_{\theta_t}(a) := \lim_{t \to \infty} \sum_{s=1}^{t} \pi_{\theta_s}(a) < \infty. \tag{29}$$

Combining Eqs. (25), (28) and (29), we have, almost surely,

$$\sup_{t \geq 1} |\theta_t(a) - \theta_1(a)| < \infty, \tag{30}$$

which implies that, almost surely,

$$\sup_{t \geq 1} |\theta_t(a)| \leq \sup_{t \geq 1} |\theta_t(a) - \theta_1(a)| + |\theta_1(a)| < \infty. \qquad \square$$

**Lemma 2** (Avoiding lack of exploration)**.** Using Algorithm 1 with any $\eta \in \Theta(1)$, there exists at least a pair of distinct actions $i, j \in [K]$ and $i \neq j$, such that, almost surely,

$$N_\infty(i) = \infty, \text{ and } N_\infty(j) = \infty. \tag{31}$$

*Proof.* First, we have, for all $t \geq 1$,

$$t = \sum_{s=1}^{t} \sum_{a \in [K]} I_s(a) \qquad \left( \sum_{a \in [K]} I_t(a) = 1 \text{ for all } t \geq 1 \right) \tag{32}$$

$$= \sum_{a \in [K]} \sum_{s=1}^{t} I_s(a) \tag{33}$$

$$= \sum_{a \in [K]} N_t(a). \tag{34}$$

By pigeonhole principle, there exists at least one action $i \in [K]$, such that, almost surely,

$$N_\infty(i) := \lim_{t \to \infty} N_t(i) = \infty. \tag{35}$$

We argue the existence of another action by contradiction. Suppose for all the other actions $j \in [K]$ and $j \neq i$, we have $N_\infty(j) < \infty$. According to the extended Borel-Cantelli Lemma 3, we have, almost surely,

$$\sum_{t=1}^{\infty} \pi_{\theta_t}(j) := \lim_{t \to \infty} \sum_{s=1}^{t} \pi_{\theta_s}(j) < \infty. \tag{36}$$

According to the update Eq. (22), we have, for all $t \geq 1$,

$$\theta_t(i) - \theta_1(i) = \sum_{s=1}^{t-1} I_s(i) \cdot \eta \cdot (1 - \pi_{\theta_s}(i)) \cdot R_s(i) + \sum_{s=1}^{t-1} (1 - I_s(i)) \cdot (-\eta) \cdot \pi_{\theta_s}(i) \cdot R_s(a_s). \tag{37}$$

By triangle inequality, we have,

$$|\theta_t(i) - \theta_1(i)| \leq \sum_{s=1}^{t-1} \left| I_s(i) \cdot \eta \cdot (1 - \pi_{\theta_s}(i)) \cdot R_s(i) \right| + \sum_{s=1}^{t-1} \left| (1 - I_s(i)) \cdot (-\eta) \cdot \pi_{\theta_s}(i) \cdot R_s(a_s) \right| \tag{38}$$

$$\leq \eta \cdot R_{\max} \cdot \sum_{s=1}^{t-1} (1 - \pi_{\theta_s}(i)) + \eta \cdot R_{\max} \cdot \sum_{s=1}^{t-1} (1 - I_s(i)) \tag{39}$$

$$= \eta \cdot R_{\max} \cdot \left( \sum_{s=1}^{t-1} \sum_{j \neq i} \pi_{\theta_s}(j) + \sum_{s=1}^{t-1} \sum_{j \neq i} I_s(j) \right) \tag{40}$$

$$= \eta \cdot R_{\max} \cdot \left( \sum_{j \neq i} \sum_{s=1}^{t-1} \pi_{\theta_s}(j) + \sum_{j \neq i} N_{t-1}(j) \right). \tag{41}$$

Combing Eqs. (36) and (38) and the assumption of $N_\infty(j) < \infty$ for all $j \neq i$, we have, almost surely,

$$\sup_{t \geq 1} |\theta_t(i)| \leq \sup_{t \geq 1} |\theta_t(i) - \theta_1(i)| + |\theta_1(i)| < \infty. \tag{42}$$

Since $N_\infty(j) < \infty$ for all $j \neq i$ by assumption, and according to Lemma 1, we have, almost surely,

$$\sup_{t \geq 1} |\theta_t(j)| < \infty. \tag{43}$$

Combining Eqs. (42) and (43), we have, for all action $a \in [K]$,

$$\sup_{t \geq 1} |\theta_t(a)| < \infty, \tag{44}$$

which implies that, there exists $c > 0$ and $c \in O(1)$, such that, for all $a \in [K]$,

$$\inf_{t \geq 1} \pi_{\theta_t}(a) = \inf_{t \geq 1} \frac{\exp\{\theta_t(a)\}}{\sum_{a' \in [K]} \exp\{\theta_t(a')\}} \geq c > 0. \tag{45}$$

Therefore, for all action $a \in [K]$,

$$\sum_{t=1}^{\infty} \pi_{\theta_t}(a) := \lim_{t \to \infty} \sum_{s=1}^{t} \pi_{\theta_s}(a) \tag{46}$$

$$\geq \lim_{t \to \infty} \sum_{s=1}^{t} c \tag{47}$$

$$= \lim_{t \to \infty} t \cdot c \tag{48}$$

$$= \infty. \tag{49}$$

According to the extended Borel-Cantelli Lemma 3, we have, almost surely, for all action $a \in [K]$,

$$N_\infty(a) = \infty, \tag{50}$$

which is a contradiction with the assumption of $N_\infty(j) < \infty$ for all the other actions $j \in [K]$ with $j \neq i$. Therefore, there exists another action $j \in [K]$ with $j \neq i$, such that $N_\infty(j) = \infty$. $\qquad \square$

**Theorem 1.** Let $K = 2$ and $r(1) > r(2)$. Using Algorithm 1 with any $\eta \in \Theta(1)$, we have, almost surely, $\pi_{\theta_t}(a^*) \to 1$ as $t \to \infty$, where $a^* := \arg\max_{a \in [K]} r(a)$ (equal to Action 1 in this case).

*Proof.* The proof uses the same notations as of [24, Theorem 5.1].

According to Lemma 2, we have, $N_\infty(1) = \infty$ and $N_\infty(2) = \infty$, i.e., both of the two actions are sampled for infinitely many times as $t \to \infty$.

Let $\mathcal{F}_t$ be the $\sigma$-algebra generated by $a_1, R_1(a_1), \cdots, a_{t-1}, R_{t-1}(a_{t-1})$:

$$\mathcal{F}_t = \sigma(\{a_1, R_1(a_1), \cdots, a_{t-1}, R_{t-1}(a_{t-1})\}). \tag{51}$$

Note that $\theta_t$ and $I_t$ (defined by Eq. (23)) are $\mathcal{F}_t$-measurable for all $t \geq 1$. Let $\mathbb{E}_t[\cdot]$ denote the conditional expectation with respect to $\mathcal{F}_t$: $\mathbb{E}_t[X] = \mathbb{E}[X|\mathcal{F}_t]$. Define the following notations,

$$W_t(a) := \theta_t(a) - \mathbb{E}_{t-1}[\theta_t(a)], \qquad (\text{"noise"}) \tag{52}$$

$$P_t(a) := \mathbb{E}_t[\theta_{t+1}(a)] - \theta_t(a). \qquad (\text{"progress"}) \tag{53}$$

For each action $a \in [K]$, for $t \geq 2$, we have the following decomposition,

$$\theta_t(a) = W_t(a) + P_{t-1}(a) + \theta_{t-1}(a). \tag{54}$$

By recursion we can determine that,

$$\theta_t(a) = \mathbb{E}[\theta_1(a)] + \sum_{s=1}^{t} W_s(a) + \sum_{s=1}^{t-1} P_s(a), \tag{55}$$

while we also have,

$$\theta_1(a) = \underbrace{\theta_1(a) - \mathbb{E}[\theta_1(a)]}_{W_1(a)} + \mathbb{E}[\theta_1(a)], \tag{56}$$

and $\mathbb{E}[\theta_1(a)]$ accounts for potential randomness in initializing $\theta_1 \in \mathbb{R}^K$. According to Proposition 1, we have,

$$P_t(a) = \mathbb{E}_t[\theta_{t+1}(a)] - \theta_t(a) = \eta \cdot \pi_{\theta_t}(a) \cdot \left(r(a) - \pi_{\theta_t}^\top r\right), \tag{57}$$

which implies that, for all $t \geq 1$,

$$P_t(a^*) > 0 > P_t(2). \tag{58}$$

Next, we show that $\sum_{s=1}^{t} P_s(a^*) \to \infty$ as $t \to \infty$ by contradiction. Suppose that,

$$\sum_{t=1}^{\infty} P_t(a^*) < \infty. \tag{59}$$

For the optimal action $a^* = 1$,

$$\sum_{s=1}^{t} P_s(a^*) = \sum_{s=1}^{t} \eta \cdot \pi_{\theta_s}(a^*) \cdot (r(a^*) - \pi_{\theta_s}^\top r) \tag{60}$$

$$= \eta \cdot \Delta \cdot \sum_{s=1}^{t} \pi_{\theta_s}(a^*) \cdot (1 - \pi_{\theta_s}(a^*)), \tag{61}$$

where $\Delta := r(a^*) - \max_{a \neq a^*} r(a) = r(1) - r(2) > 0$ is the reward gap. Denote that, for all $t \geq 1$,

$$V_t(a^*) := \frac{5}{18} \cdot \sum_{s=1}^{t-1} \pi_{\theta_s}(a^*) \cdot (1 - \pi_{\theta_s}(a^*)). \tag{62}$$

According to Lemma 8 (using Eqs. (58) to (60) and (62)), we have, almost surely,

$$\sup_{t \geq 1} |\theta_t(a^*)| < \infty. \tag{63}$$

For the sub-optimal action (Action 2), we have,

$$\sum_{s=1}^{t} P_s(2) = \sum_{s=1}^{t} \eta \cdot \pi_{\theta_s}(2) \cdot (r(2) - \pi_{\theta_s}^\top r) \tag{64}$$

$$= -\eta \cdot \Delta \cdot \sum_{s=1}^{t} \pi_{\theta_s}(2) \cdot (1 - \pi_{\theta_s}(2)) \tag{65}$$

$$= -\sum_{s=1}^{t} P_s(a^*), \tag{66}$$

and also denote, for all $t \geq 1$,

$$V_t(2) := \frac{5}{18} \cdot \sum_{s=1}^{t-1} \pi_{\theta_s}(2) \cdot (1 - \pi_{\theta_s}(2)) = V_t(a^*). \tag{67}$$

According to Lemma 8 (using Eqs. (58), (59), (64) and (67)), we have, almost surely,

$$\sup_{t \geq 1} |\theta_t(2)| < \infty. \tag{68}$$

Combining Eqs. (63) and (68), there exists $c > 0$ and $c \in O(1)$, such that, for all $a \in \{a^*, 2\}$,

$$\inf_{t \geq 1} \pi_{\theta_t}(a) = \inf_{t \geq 1} \frac{\exp\{\theta_t(a)\}}{\sum_{a' \in [K]} \exp\{\theta_t(a')\}} \geq c > 0, \tag{69}$$

which implies that,

$$\sum_{t=1}^{\infty} \pi_{\theta_t}(a^*) \cdot (1 - \pi_{\theta_t}(a^*)) := \lim_{t \to \infty} \sum_{s=1}^{t} \pi_{\theta_s}(a^*) \cdot (1 - \pi_{\theta_s}(a^*)) \tag{70}$$

$$\geq \lim_{t \to \infty} \sum_{s=1}^{t} c \cdot c \tag{71}$$

$$= \lim_{t \to \infty} t \cdot c^2 \tag{72}$$

$$= \infty, \tag{73}$$

which is a contradiction with the assumption of Eq. (59). Therefore, we have,

$$\sum_{s=1}^{t} P_s(a^*) \to \infty, \text{ as } t \to \infty. \tag{74}$$

According to Lemma 9 (using Eqs. (58), (60), (62) and (74)), we have, almost surely,

$$\theta_t(a^*) \to \infty, \text{ as } t \to \infty. \tag{75}$$

Similarly, according to Lemma 10 (using Eqs. (58), (64), (67) and (74)), we have, almost surely,

$$\lim_{t \to \infty} \theta_t(2) = -\infty. \tag{76}$$

Note that,

$$\pi_{\theta_t}(a^*) = \frac{\pi_{\theta_t}(a^*)}{\pi_{\theta_t}(2) + \pi_{\theta_t}(a^*)} \tag{77}$$

$$= \frac{1}{\exp\{\theta_t(2) - \theta_t(a^*)\} + 1}. \tag{78}$$

Combining Eqs. (75) to (77), we have, almost surely,

$$\pi_{\theta_t}(a^*) \to 1, \text{ as } t \to \infty. \qquad \square$$

**Theorem 2.** Under Assumption 1, given $K \geq 2$, using Algorithm 1 with any $\eta \in \Theta(1)$, we have, almost surely, $\pi_{\theta_t}(a^*) \to 1$ as $t \to \infty$, where $a^* = \arg\max_{a \in [K]} r(a)$ is the optimal action.

*Proof.* Similar to the proof of Theorem 1, we define $\mathcal{F}_t$ to be the $\sigma$-algebra generated by $a_1, R_1(a_1)$, $\cdots, a_{t-1}, R_{t-1}(a_{t-1})$ i.e.

$$\mathcal{F}_t = \sigma(\{a_1, R_1(a_1), \cdots, a_{t-1}, R_{t-1}(a_{t-1})\}). \tag{79}$$

Note that $\theta_t$ and $I_t$ (defined by Eq. (23)) are $\mathcal{F}_t$-measurable for all $t \geq 1$. Let $\mathbb{E}_t[\cdot]$ denote the conditional expectation with respect to $\mathcal{F}_t$: $\mathbb{E}_t[X] = \mathbb{E}[X|\mathcal{F}_t]$. Define the following notations,

$$W_t(a) := \theta_t(a) - \mathbb{E}_{t-1}[\theta_t(a)], \qquad (\text{"noise"}) \tag{80}$$
$$P_t(a) := \mathbb{E}_t[\theta_{t+1}(a)] - \theta_t(a). \qquad (\text{"progress"}) \tag{81}$$

For each action $a \in [K]$, for $t \geq 2$, we have the following decomposition,

$$\theta_t(a) = W_t(a) + P_{t-1}(a) + \theta_{t-1}(a). \tag{82}$$

By recursion we can determine that,

$$\theta_t(a) = \mathbb{E}[\theta_1(a)] + \sum_{s=1}^{t} W_s(a) + \sum_{s=1}^{t-1} P_s(a), \tag{83}$$

while we also have,

$$\theta_1(a) = \underbrace{\theta_1(a) - \mathbb{E}[\theta_1(a)]}_{W_1(a)} + \mathbb{E}[\theta_1(a)], \tag{84}$$

and $\mathbb{E}[\theta_1(a)]$ accounts for potential randomness in initializing $\theta_1 \in \mathbb{R}^K$. For an action $a \in [K]$,

$$\mathcal{A}^+(a) := \{a^+ \in [K] : r(a^+) > r(a)\}, \tag{85}$$
$$\mathcal{A}^-(a) := \{a^- \in [K] : r(a^-) < r(a)\}. \tag{86}$$

Define $\mathcal{A}_\infty$ as the set of actions which are sampled for infinitely many times as $t \to \infty$, i.e.,

$$\mathcal{A}_\infty := \{a \in [K] \mid N_\infty(a) = \infty\}. \tag{87}$$

**First part.** We show that $N_\infty(a^*) = \infty$ by contradiction.

Suppose $a^* \notin \mathcal{A}_\infty$ i.e. $N_\infty(a^*) < \infty$. According to Lemma 2, we have, $|\mathcal{A}_\infty| \geq 2$. Since $a^* \notin \mathcal{A}_\infty$ by assumption, there must be at least two other sub-optimal actions $i_1, i_2 \in [K]$, $i_1 \neq i_2$, such that,

$$N_\infty(i_1) = N_\infty(i_2) = \infty. \tag{88}$$

In particular, define

$$i_1 := \underset{\substack{a \in [K], \\ N_\infty(a) = \infty}}{\arg\min} r(a), \tag{89}$$

$$i_2 := \underset{\substack{a \in [K], \\ N_\infty(a) = \infty}}{\arg\max} r(a). \tag{90}$$

Using the update for arm $i_1$, we know that, for all $s \geq 1$,

$$P_s(i_1) = \eta \cdot \pi_{\theta_s}(i_1) \cdot (r(i_1) - \pi_{\theta_s}^\top r) \tag{91}$$

By definition and because of Assumption 1, we have that $r(i_1) < r(i_2) < r(a^*)$. Furthermore, according to Lemma 6, we have, for sufficiently large $\tau \geq 1$,

$$r(i_1) < \pi_{\theta_t}^\top r < r(i_2), \tag{92}$$

which implies that after some large enough $\tau \geq 1$,

$$\sum_{s=\tau}^{t} P_s(i_1) = \sum_{s=\tau}^{t} \eta \cdot \pi_{\theta_s}(i_1) \cdot (r(i_1) - \pi_{\theta_s}^\top r) \tag{93}$$

$$< 0. \quad \text{(by Eq. (92))} \tag{94}$$

$$r(i_1) - \pi_{\theta_s}^\top r = \sum_{a \neq i_1} \pi_{\theta_s}(a) \left[ r(i_1) - r(a) \right]$$

$$= - \sum_{a^+ \in \mathcal{A}^+(i_1)} \pi_{\theta_s}(a^+) \cdot (r(a^+) - r(i_1)) + \sum_{a^- \in \mathcal{A}^-(i_1)} \pi_{\theta_s}(a^-) \cdot (r(i_1) - r(a^-)). \tag{95}$$

From Eq. (89), we have,

$$\mathcal{A}_\infty \subseteq \mathcal{A}^+(i_1) \cup \{i_1\}, \tag{96}$$

which implies that, for all $a^- \in \mathcal{A}^-(i_1)$, we have,

$$N_\infty(a^-) < \infty. \tag{97}$$

Note that, by definition, $i_2 \in \mathcal{A}^+(i_1)$, and

$$N_\infty(i_2) = \infty. \tag{98}$$

In order to bound Eq. (95) using the above relations, note that,

$$\sum_{a^- \in \mathcal{A}^-(i_1)} \frac{\pi_{\theta_s}(a^-)}{\sum_{a^+ \in \mathcal{A}^+(i_1)} \pi_{\theta_s}(a^+) \cdot (r(a^+) - r(i_1))} \cdot (r(i_1) - r(a^-)) \tag{99}$$

$$< \sum_{a^- \in \mathcal{A}^-(i_1)} \frac{\pi_{\theta_s}(a^-)}{\pi_{\theta_s}(i_2) \cdot (r(i_2) - r(i_1))} \cdot (r(i_1) - r(a^-))$$

$$\text{(Fewer terms in the denominator)}$$

By Lemma 5, for large enough $\tau \geq 1$, for all $a^-$, $\frac{\pi_{\theta_s}(a^-)}{\pi_{\theta_s}(i_2)} \leq \frac{1}{|\mathcal{A}^-(i_1)|} \cdot \frac{r(i_2)-r(i_1)}{r(i_1)-r(a^-)}$. Hence,

$$\leq \sum_{a^- \in \mathcal{A}^-(i_1)} \frac{1}{2} \cdot \frac{1}{|\mathcal{A}^-(i_1)|} \cdot \frac{r(i_2) - r(i_1)}{r(i_1) - r(a^-)} \cdot \frac{r(i_1) - r(a^-)}{r(i_2) - r(i_1)} \tag{100}$$

$$= \frac{1}{2} \tag{101}$$

$$\implies \sum_{a^- \in \mathcal{A}^-(i_1)} \pi_{\theta_s}(a^-) \cdot (r(i_1) - r(a^-)) \leq \frac{1}{2} \sum_{a^+ \in \mathcal{A}^+(i_1)} \pi_{\theta_s}(a^+) \cdot (r(a^+) - r(i_1)). \tag{102}$$

Combining Eqs. (94), (95) and (102), we have,

$$\sum_{s=\tau}^{t} P_s(i_1) = \sum_{s=\tau}^{t} \eta \cdot \pi_{\theta_s}(i_1) \cdot (r(i_1) - \pi_{\theta_s}^\top r) \tag{103}$$

$$\leq -\frac{\eta}{2} \cdot \sum_{s=\tau}^{t} \pi_{\theta_s}(i_1) \sum_{a^+ \in \mathcal{A}^+(i_1)} \pi_{\theta_s}(a^+) \cdot (r(a^+) - r(i_1)) \tag{104}$$

$$\leq -\frac{\eta \cdot \Delta}{2} \cdot \sum_{s=\tau}^{t} \pi_{\theta_s}(i_1) \sum_{a^+ \in \mathcal{A}^+(i_1)} \pi_{\theta_s}(a^+), \tag{105}$$

where

$$\Delta := \min_{i,j \in [K], \ i \neq j} |r(i) - r(j)| > 0. \qquad \text{(by Assumption 1)} \tag{106}$$

Bounding the variance for arm $i_1$, we have that for all large enough $\tau \geq 1$,

$$V_t(i_1) := \frac{5}{18} \cdot \sum_{s=\tau}^{t-1} \pi_{\theta_s}(i_1) \cdot (1 - \pi_{\theta_s}(i_1)) \tag{107}$$

$$= \frac{5}{18} \cdot \sum_{s=\tau}^{t-1} \pi_{\theta_s}(i_1) \cdot \left( \sum_{a^- \in \mathcal{A}^-(i_1)} \pi_{\theta_s}(a^-) + \sum_{a^+ \in \mathcal{A}^+(i_1)} \pi_{\theta_s}(a^+) \right) \tag{108}$$

Using similar calculations as for the progress term, we have,

$$\sum_{a^- \in \mathcal{A}^-(i_1)} \frac{\pi_{\theta_s}(a^-)}{\sum_{a^+ \in \mathcal{A}^+(i_1)} \pi_{\theta_s}(a^+)} < \sum_{a^- \in \mathcal{A}^-(i_1)} \frac{\pi_{\theta_s}(a^-)}{\pi_{\theta_s}(i_2)} \qquad \text{(Fewer terms in the denominator)}$$

By Lemma 5, for large enough $\tau \geq 1$, for all $a^-$, $\frac{\pi_{\theta_s}(a^-)}{\pi_{\theta_s}(i_2)} \leq \frac{13}{5} \frac{1}{|\mathcal{A}^-(i_1)|}$. Hence,

$$\leq \sum_{a^- \in \mathcal{A}^-(i_1)} \frac{13}{5} \cdot \frac{1}{|\mathcal{A}^-(i_1)|} \tag{109}$$

$$= \frac{13}{5} \tag{110}$$

$$\implies \sum_{a^- \in \mathcal{A}^-(i_1)} \pi_{\theta_s}(a^-) \leq \frac{13}{5} \sum_{a^+ \in \mathcal{A}^+(i_1)} \pi_{\theta_s}(a^+) \tag{111}$$

Combining Eqs. (108) and (111), we have that for large enough $\tau \geq 1$,

$$V_t(i_1) = \frac{5}{18} \cdot \sum_{s=\tau}^{t-1} \pi_{\theta_s}(i_1) \cdot (1 - \pi_{\theta_s}(i_1)) \tag{112}$$

$$\leq \sum_{s=\tau}^{t-1} \pi_{\theta_s}(i_1) \sum_{a^+ \in \mathcal{A}^+(i_1)} \pi_{\theta_s}(a^+). \tag{113}$$

According to Lemma 12 (using Eqs. (94), (105) and (113)), we have, almost surely,

$$\sup_{t \geq 1} \theta_t(i_1) < \infty. \tag{114}$$

Since $N_\infty(a^*) < \infty$ by assumption, and according to Lemma 1, we have, almost surely,

$$\inf_{t \geq 1} \theta_t(a^*) > -\infty. \tag{115}$$

Combining Eqs. (114) and (115), we have,

$$\sup_{t \geq 1} \frac{\pi_{\theta_t}(i_1)}{\pi_{\theta_t}(a^*)} = \sup_{t \geq 1} \exp\{\theta_t(i_1) - \theta_t(a^*)\} < \infty. \tag{116}$$

On the other hand, by Lemma 5, we have,

$$\sup_{t \geq 1} \frac{\pi_{\theta_t}(i_1)}{\pi_{\theta_t}(a^*)} = \infty \tag{117}$$

which contradicts Eq. (116). Hence, the assumption that $N_\infty(a^*) < \infty$ cannot hold. This completes the proof by contradiction, and implies that $N_\infty(a^*) = \infty$.

**Second part.** With $a^* \in \mathcal{A}_\infty$ i.e. $N_\infty(a^*) = \infty$, we now argue that $\pi_{\theta_t}(a^*) \to 1$ as $t \to \infty$ almost surely.

According to Lemma 2, we have, $|\mathcal{A}_\infty| \geq 2$. Since $a^* \in \mathcal{A}_\infty$ by assumption, there must be at least one sub-optimal action $i_1 \in [K]$ with $r(i_1) < r(a^*)$, such that,

$$N_\infty(i_1) = \infty. \tag{118}$$

In particular, define

$$i_1 := \arg\min_{\substack{a \in [K], \\ N_\infty(a) = \infty}} r(a). \tag{119}$$

According to Lemma 6, we have, for all sufficiently large $\tau \geq 1$,

$$r(i_1) < \pi_{\theta_t}^\top r < r(a^*). \tag{120}$$

Using the same arguments as in the first part (except that $i_2$ in Eq. (98) is replaced with $a^*$), both Eqs. (105) and (113) hold.

Next, we argue that $\sum_{s=\tau}^t \pi_{\theta_s}(i_1) \sum_{a^+ \in \mathcal{A}^+(i_1)} \pi_{\theta_s}(a^+) \to \infty$ as $t \to \infty$ by contradiction.

Suppose

$$\sum_{t=\tau}^\infty \pi_{\theta_t}(i_1) \sum_{a^+ \in \mathcal{A}^+(i_1)} \pi_{\theta_t}(a^+) < \infty. \tag{121}$$

According to Lemma 8 (using Eqs. (94), (105), (113) and (121)), we have,

$$\sup_{t \geq 1} |\theta_t(i_1)| < \infty. \tag{122}$$

Calculating the progress and variance for arm $a^*$, for $t \geq 1$,

$$P_t(a^*) = \eta \cdot \pi_{\theta_t}(a^*) \cdot (r(a^*) - \pi_{\theta_t}^\top r) \tag{123}$$

$$\geq \eta \cdot \Delta \cdot \pi_{\theta_t}(a^*) \cdot \left(1 - \pi_{\theta_t}(a^*)\right). \qquad \left(\Delta := r(a^*) - \max_{a \neq a^*} r(a)\right) \tag{124}$$

$$\geq 0. \tag{125}$$

Denote that, for all $t \geq 1$,

$$V_t(a^*) := \frac{5}{18} \cdot \sum_{s=1}^{t-1} \pi_{\theta_s}(a^*) \cdot (1 - \pi_{\theta_s}(a^*)). \tag{126}$$

According to Lemma 11 (using Eqs. (123) and (126)), we have,

$$\inf_{t \geq 1} \theta_t(a^*) > -\infty. \tag{127}$$

Combining Eqs. (122) and (127), we have,

$$\sup_{t \geq 1} \frac{\pi_{\theta_t}(i_1)}{\pi_{\theta_t}(a^*)} = \sup_{t \geq 1} \exp\{\theta_t(i_1) - \theta_t(a^*)\} < \infty, \tag{128}$$

For all $t \geq 1$, $|\theta_t(i_1)| < \infty$ and since there is at least one arm $(a^*)$ s.t. $\inf_{t \geq 1} \theta_t(a) > -\infty$, there exists $\epsilon > 0$ and $\epsilon \in O(1)$, such that,

$$\sup_{t \geq 1} \pi_{\theta_t}(i_1) < 1 - 2\,\epsilon. \tag{129}$$

According to Eq. (118), we know that $N_\infty(i_1) = \infty$ and for all $a^- \in \mathcal{A}^-(i_1)$, $N_\infty(a^-) < \infty$. Using Lemma 5, we have, for all large enough $t \geq 1$, $\frac{\pi_{\theta_t}(a^-)}{\pi_{\theta_t}(i_1)} < \frac{\epsilon}{|\mathcal{A}^-(i_1)|}$.

$$\pi_{\theta_t}(i_1) + \sum_{a^+ \in \mathcal{A}^+(i_1)} \pi_{\theta_t}(a^+) = 1 - \pi_{\theta_t}(i_1) \sum_{a^- \in \mathcal{A}^-(i_1)} \frac{\pi_{\theta_t}(a^-)}{\pi_{\theta_t}(i_1)} > 1 - \pi_{\theta_t}(i_1)\,\epsilon \tag{130}$$

$$\implies \pi_{\theta_t}(i_1) + \sum_{a^+ \in \mathcal{A}^+(i_1)} \pi_{\theta_t}(a^+) \geq 1 - \epsilon. \tag{131}$$

Combining Eqs. (129) and (131), we have, for all large enough $t \geq 1$,

$$\sum_{a^+ \in \mathcal{A}^+(i_1)} \pi_{\theta_t}(a^+) \geq \epsilon, \tag{132}$$

which implies that,

$$\sum_{s=\tau}^{\infty} \pi_{\theta_s}(i_1) \sum_{a^+ \in \mathcal{A}^+(i_1)} \pi_{\theta_s}(a^+) \geq \epsilon \cdot \sum_{s=\tau}^{\infty} \pi_{\theta_s}(i_1) \tag{133}$$

$$= \infty, \qquad \text{(since } N_{\infty}(i_1) = \infty \text{ and by Lemma 3)} \tag{134}$$

which contradicts the assumption of Eq. (121). This completes the proof by contradiction, and therefore, we have,

$$\sum_{s=\tau}^{t} \pi_{\theta_s}(i_1) \sum_{a^+ \in \mathcal{A}^+(i_1)} \pi_{\theta_s}(a^+) \to \infty, \text{ as } t \to \infty. \tag{135}$$

According to Lemma 10 (using Eqs. (105), (113) and (135)), we have, almost surely,

$$\theta_t(i_1) \to -\infty, \text{ as } t \to \infty. \tag{136}$$

Combining Eqs. (127) and (136), we have, almost surely,

$$\frac{\pi_{\theta_t}(a^*)}{\pi_{\theta_t}(i_1)} = \exp\{\theta_t(a^*) - \theta_t(i_1)\} \to \infty, \text{ as } t \to \infty. \tag{137}$$

Hence, we have proved that if $N_{\infty}(i_1) = \infty$ and $r(i_1) < \pi_{\theta_t}^{\top} r < r(a^*)$ for sufficiently large $\tau \geq 1$, then, $\frac{\pi_{\theta_t}(a^*)}{\pi_{\theta_t}(i_1)} \to \infty$, as $t \to \infty$.

In order to use this argument recursively, consider sorting the action indices in $\mathcal{A}_{\infty}$ according to their descending expected reward values,

$$r(a^*) > r(i_{|\mathcal{A}_{\infty}|-1}) > r(i_{|\mathcal{A}_{\infty}|-2}) > \cdots > r(i_2) > r(i_1). \tag{138}$$

We know that,

$$\pi_{\theta_t}^{\top} r - r(i_2) = \sum_{a \neq i_2} \pi_{\theta_t}(a) \cdot (r(a) - r(i_2)) \tag{139}$$

$$= \sum_{a^- \in \mathcal{A}^-(i_2)} \pi_{\theta_t}(a^-) \cdot (r(a^-) - r(i_2)) + \sum_{a^+ \in \mathcal{A}^+(i_2)} \pi_{\theta_t}(a^+) \cdot (r(a^+) - r(i_2)) \tag{140}$$

$$> \pi_{\theta_t}(a^*) \cdot (r(a^*) - r(i_2)) - \sum_{a^- \in \mathcal{A}^-(i_2)} \pi_{\theta_t}(a^-) \cdot (r(i_2) - r(a^-)) \tag{141}$$

$$= \pi_{\theta_t}(a^*) \cdot \left[ r(a^*) - r(i_2) - \sum_{a^- \in \mathcal{A}^-(i_2)} \frac{\pi_{\theta_t}(a^-)}{\pi_{\theta_t}(a^*)} \cdot (r(i_2) - r(a^-)) \right], \tag{142}$$

According to Lemma 5, for all $a^- \in \mathcal{A}^-(i_2)$ with $a^- \neq i_1$, we have,

$$\frac{\pi_{\theta_t}(a^*)}{\pi_{\theta_t}(a^-)} \to \infty, \text{ as } t \to \infty. \tag{143}$$

Combining Eqs. (137) and (143), we have, for all $a^- \in \mathcal{A}^-(i_2)$,

$$\frac{\pi_{\theta_t}(a^*)}{\pi_{\theta_t}(a^-)} \to \infty, \text{ as } t \to \infty. \tag{144}$$

Hence, for all sufficiently large $\tau \geq 1$, for all $a^- \in \mathcal{A}^-(i_2)$,

$$\frac{\pi_{\theta_t}(a^-)}{\pi_{\theta_t}(a^*)} \leq \frac{1}{2 |\mathcal{A}^-(i_2)|} \frac{r(a^*) - r(i_2)}{r(i_2) - r(a^-)}. \tag{145}$$

Combining Eqs. (142) and (145), we have, for all sufficiently large $t \geq 1$,

$$\pi_{\theta_t}^\top r - r(i_2) > \pi_{\theta_t}(a^*) \cdot \frac{r(a^*) - r(i_2)}{2} > 0. \tag{146}$$

Hence we have, for all sufficiently large $\tau \geq 1$,

$$r(i_2) < \pi_{\theta_t}^\top r < r(a^*) \tag{147}$$

Comparing Eqs. (120) and (147), we can use a similar argument for $i_2$ and conclude that, for sufficiently large $\tau \geq 1$, then, $\frac{\pi_{\theta_t}(a^*)}{\pi_{\theta_t}(i_2)} \to \infty$, as $t \to \infty$. This further implies that

$$r(i_3) < \pi_{\theta_t}^\top r < r(a^*). \tag{148}$$

Continuing this recursive argument, we have, for all actions $a \in \mathcal{A}_\infty$ with $a \neq a^*$,

$$\frac{\pi_{\theta_t}(a^*)}{\pi_{\theta_t}(a)} \to \infty, \text{ as } t \to \infty. \tag{149}$$

Meanwhile, according to Lemma 5, we have, for all actions $a \notin \mathcal{A}_\infty$,

$$\frac{\pi_{\theta_t}(a^*)}{\pi_{\theta_t}(a)} \to \infty, \text{ as } t \to \infty. \tag{150}$$

Combining Eqs. (149) and (150), we have, for all sub-optimal actions $a \in [K]$ with $r(a) < r(a^*)$,

$$\frac{\pi_{\theta_t}(a^*)}{\pi_{\theta_t}(a)} \to \infty, \text{ as } t \to \infty. \tag{151}$$

Finally, note that,

$$\pi_{\theta_t}(a^*) = \frac{\pi_{\theta_t}(a^*)}{\sum_{a \in [K]: \, r(a) < r(a^*)} \pi_{\theta_t}(a) + \pi_{\theta_t}(a^*)} \tag{152}$$

$$= \frac{1}{\sum_{a \in [K]: \, r(a) < r(a^*)} \frac{\pi_{\theta_t}(a)}{\pi_{\theta_t}(a^*)} + 1}. \tag{153}$$

Combining Eqs. (151) and (153), we have, almost surely,

$$\pi_{\theta_t}(a^*) \to 1, \text{ as } t \to \infty. \qquad \square$$

## B   Miscellaneous Extra Supporting Results

**Lemma 3** (Extended Borel-Cantelli Lemma, Corollary 5.29 of [5]). *Let $(\mathcal{F}_n)_{n \geq 1}$ be a filtration, $A_n \in \mathcal{F}_n$. Then, almost surely,*

$$\{\omega \,:\, \omega \in A_n \text{ infinitely often }\} = \left\{\omega \,:\, \sum_{n=1}^{\infty} \mathbb{P}(A_n | \mathcal{F}_n) = \infty\right\}. \tag{154}$$

**Lemma 4** (Freedman's inequality [10, 7], Theorem C.3 of [24]). *Let $X_1, X_2, \ldots$ be a sequence of random variables, such that for all $t \geq 1$, $|X_t| \leq 1/2$. Define*

$$S_n := \left|\sum_{t=1}^{n} \mathbb{E}[X_t | X_1, \ldots, X_{t-1}] - X_t\right| \quad and \quad V_n := \sum_{t=1}^{n} \mathrm{Var}[X_t | X_1, \ldots, X_{t-1}]. \tag{155}$$

*Then, for all $\delta > 0$,*

$$\Pr\left(\exists\, n :\, S_n \geq 6\sqrt{\left(V_n + \frac{4}{3}\right) \log\left(\frac{V_n + 1}{\delta}\right)} + 2\log\left(\frac{1}{\delta}\right) + \frac{4}{3}\log 3\right) \leq \delta. \tag{156}$$

**Lemma 5.** *Using Algorithm 1, for any two different actions $i, j \in [K]$ with $i \neq j$, if $N_\infty(i) = \infty$ and $N_\infty(j) < \infty$, then we have, almost surely,*

$$\sup_{t \geq 1} \frac{\pi_{\theta_t}(i)}{\pi_{\theta_t}(j)} = \infty. \tag{157}$$

*Proof.* We prove the result by contradiction. Suppose

$$c := \sup_{t \geq 1} \frac{\pi_{\theta_t}(i)}{\pi_{\theta_t}(j)} < \infty. \tag{158}$$

According to the extended Borel-Cantelli Lemma 3, we have, for all $a \in [K]$, almost surely,

$$\left\{\sum_{t \geq 1} \pi_{\theta_t}(j) = \infty\right\} = \{N_\infty(j) = \infty\}. \tag{159}$$

Hence, taking complements, we have,

$$\left\{\sum_{t \geq 1} \pi_{\theta_t}(j) < \infty\right\} = \{N_\infty(j) < \infty\} \tag{160}$$

also holds almost surely, which implies that,

$$\sum_{t=1}^{\infty} \pi_{\theta_t}(i) = \infty, \text{ and} \tag{161}$$

$$\sum_{t=1}^{\infty} \pi_{\theta_t}(j) < \infty. \tag{162}$$

Therefore, we have,

$$\sum_{t=1}^{\infty} \pi_{\theta_t}(i) = \sum_{t=1}^{\infty} \pi_{\theta_t}(j) \cdot \frac{\pi_{\theta_t}(i)}{\pi_{\theta_t}(j)} \tag{163}$$

$$\leq c \cdot \sum_{t=1}^{\infty} \pi_{\theta_t}(j) \tag{164}$$

$$< \infty, \tag{165}$$

which is a contradiction with Eq. (161). $\qquad\square$

**Lemma 6.** *Using Algorithm 1, we have, almost surely, for all large enough $t \geq 1$,*

$$r(i_1) < \pi_{\theta_t}^\top r < r(i_2), \tag{166}$$

*where $i_1, i_2 \in [K]$ and $i_1 \neq i_2$ are the action indices defined as,*

$$i_1 := \underset{\substack{a \in [K], \\ N_\infty(a) = \infty}}{\arg\min} \; r(a), \tag{167}$$

$$i_2 := \underset{\substack{a \in [K], \\ N_\infty(a) = \infty}}{\arg\max} \; r(a). \tag{168}$$

*Proof.* Define $\mathcal{A}_\infty$ as the set of actions which are sampled for infinitely many times as $t \to \infty$, i.e.,

$$\mathcal{A}_\infty := \{a \in [K] \mid N_\infty(a) = \infty\}. \tag{169}$$

According to Lemma 2, we have $|\mathcal{A}_\infty| \geq 2$, which implies that $i_1 \neq i_2$.

Given any sub-optimal action $i \in [K]$ with $r(i) < r(a^*)$, we partition the remaining actions into two parts using $r(i)$.

$$\mathcal{A}^+(i) := \left\{ a^+ \in [K] : r(a^+) > r(i) \right\}, \tag{170}$$

$$\mathcal{A}^-(i) := \left\{ a^- \in [K] : r(a^-) < r(i) \right\}. \tag{171}$$

By definition, we have,

$$\mathcal{A}_\infty \subseteq \mathcal{A}^+(i_1) \cup \{i_1\}, \text{ and} \tag{172}$$

$$\mathcal{A}_\infty \subseteq \mathcal{A}^-(i_2) \cup \{i_2\}. \tag{173}$$

**First part.** $r(i_1) < \pi_{\theta_t}^\top r$.

If $r(i_1) = \min_{a \in [K]} r(a)$, i.e., $i_1$ is the "worst action", then $r(i_1) < \pi_{\theta_t}^\top r$ holds trivially. Otherwise, suppose $r(i_1) \neq \min_{a \in [K]} r(a)$. We have,

$$\pi_{\theta_t}^\top r - r(i_1) = \sum_{a^+ \in \mathcal{A}^+(i_1)} \pi_{\theta_t}(a^+) \cdot (r(a^+) - r(i_1)) - \sum_{a^- \in \mathcal{A}^-(i_1)} \pi_{\theta_t}(a^-) \cdot (r(i_1) - r(a^-)). \tag{174}$$

Consider the non-empty set $\mathcal{A}_\infty \cap \mathcal{A}^+(i_1)$. Pick an action $j_1 \in \mathcal{A}_\infty \cap \mathcal{A}^+(i_1)$, and ignore all the other actions $a^+ \in \mathcal{A}^+(i_1)$ with $a^+ \neq j_1$ in the above equation. We have,

$$\pi_{\theta_t}^\top r - r(i_1) > \pi_{\theta_t}(j_1) \cdot (r(j_1) - r(i_1)) - \sum_{a^- \in \mathcal{A}^-(i_1)} \pi_{\theta_t}(a^-) \cdot (r(i_1) - r(a^-)) \tag{175}$$

$$= \pi_{\theta_t}(j_1) \cdot \left[ r(j_1) - r(i_1) - \sum_{a^- \in \mathcal{A}^-(i_1)} \frac{\pi_{\theta_t}(a^-)}{\pi_{\theta_t}(j_1)} \cdot (r(i_1) - r(a^-)) \right], \tag{176}$$

where the first inequality is because of $r(a^+) - r(i_1) > 0$ for all $a^+ \in \mathcal{A}^+(i_1)$ by Eq. (170). Note that $N_\infty(j_1) = \infty$ and $N_\infty(a^-) < \infty$. According to Lemma 5, we have, for all large enough $t \geq 1$,

$$\sum_{a^- \in \mathcal{A}^-(i_1)} \frac{\pi_{\theta_t}(a^-)}{\pi_{\theta_t}(j_1)} \cdot (r(i_1) - r(a^-)) = \sum_{a^- \in \mathcal{A}^-(i_1)} (r(i_1) - r(a^-)) \Big/ \frac{\pi_{\theta_t}(j_1)}{\pi_{\theta_t}(a^-)} \tag{177}$$

$$< \sum_{a^- \in \mathcal{A}^-(i_1)} (r(i_1) - r(a^-)) \cdot \frac{1}{|\mathcal{A}^-(i_1)|} \cdot \frac{r(j_1) - r(i_1)}{r(i_1) - r(a^-)} \cdot \frac{1}{2} \tag{178}$$

$$= \frac{r(j_1) - r(i_1)}{2}. \tag{179}$$

Combining Eqs. (175) and (177), we have,

$$\pi_{\theta_t}^\top r - r(i_1) > \pi_{\theta_t}(j_1) \cdot \frac{r(j_1) - r(i_1)}{2} > 0, \tag{180}$$

where the last inequality is because $j_1 \in \mathcal{A}^+(i_1)$.

**Second part.** $\pi_{\theta_t}^\top r < r(i_2)$.

The arguments are similar to the first part. If $r(i_2) = r(a^*)$, i.e., $i_2$ is the optimal action, then $\pi_{\theta_t}^\top r < r(i_2)$ holds trivially. Otherwise, suppose $r(i_2) \neq r(a^*)$. We have,

$$r(i_2) - \pi_{\theta_t}^\top r = \sum_{a^- \in \mathcal{A}^-(i_2)} \pi_{\theta_t}(a^-) \cdot (r(i_2) - r(a^-)) - \sum_{a^+ \in \mathcal{A}^+(i_2)} \pi_{\theta_t}(a^+) \cdot (r(a^+) - r(i_2)). \tag{181}$$

Consider the non-empty set $\mathcal{A}_\infty \cap \mathcal{A}^-(i_2)$. Pick an action $j_2 \in \mathcal{A}_\infty \cap \mathcal{A}^-(i_2)$, and ignore all the other actions $a^- \in \mathcal{A}^-(i_2)$ with $a^- \neq j_2$ in the above equation. We have,

$$r(i_2) - \pi_{\theta_t}^\top r \geq \pi_{\theta_t}(j_2) \cdot (r(i_2) - r(j_2)) - \sum_{a^+ \in \mathcal{A}^+(i_2)} \pi_{\theta_t}(a^+) \cdot (r(a^+) - r(i_2)) \tag{182}$$

$$= \pi_{\theta_t}(j_2) \cdot \left[ r(i_2) - r(j_2) - \sum_{a^+ \in \mathcal{A}^+(i_2)} \frac{\pi_{\theta_t}(a^+)}{\pi_{\theta_t}(j_2)} \cdot (r(a^+) - r(i_2)) \right], \tag{183}$$

where the first inequality is because of $r(i_2) - r(a^-) > 0$ for all $a^- \in \mathcal{A}^-(i_2)$ by Eq. (170). Note that $N_\infty(j_2) = \infty$ and $N_\infty(a^+) < \infty$. According to Lemma 5, we have, for all large enough $t \geq 1$,

$$\sum_{a^+ \in \mathcal{A}^+(i_2)} \frac{\pi_{\theta_t}(a^+)}{\pi_{\theta_t}(j_2)} \cdot (r(a^+) - r(i_2)) = \sum_{a^+ \in \mathcal{A}^+(i_2)} (r(a^+) - r(i_2)) \Big/ \frac{\pi_{\theta_t}(j_2)}{\pi_{\theta_t}(a^+)} \tag{184}$$

$$< \sum_{a^+ \in \mathcal{A}^+(i_2)} (r(a^+) - r(i_2)) \cdot \frac{1}{|\mathcal{A}^+(i_2)|} \cdot \frac{r(i_2) - r(j_2)}{r(a^+) - r(i_2)} \cdot \frac{1}{2} \tag{185}$$

$$= \frac{r(i_2) - r(j_2)}{2}. \tag{186}$$

Combining Eqs. (182) and (184), we have,

$$r(i_2) - \pi_{\theta_t}^\top r \geq \pi_{\theta_t}(j_2) \cdot \frac{r(i_2) - r(j_2)}{2} > 0, \tag{187}$$

where the last inequality is because $j_2 \in \mathcal{A}^-(i_2)$. $\qquad\square$

For the following lemmas, we will use the notation defined in the proofs for Theorems 1 and 2.

**Lemma 7** (Concentration of noise). *Given an action $a \in [K]$. We have, with probability at least $1 - \delta$,*

$$\forall t: \quad \left| \sum_{s=1}^t W_{s+1}(a) \right| \leq 36\, \eta\, R_{\max} \sqrt{(V_t(a) + 4/3) \log\left( \frac{V_t(a) + 1}{\delta} \right)} + 12\, \eta\, R_{\max}\, \log(1/\delta) + 8\, \eta\, R_{\max} \log 3, \tag{188}$$

*where*

$$V_t(a) := \frac{5}{18} \cdot \sum_{s=1}^t \pi_{\theta_s}(a) \cdot (1 - \pi_{\theta_s}(a)), \tag{189}$$

*and*

$$W_t(a) := \theta_t(a) - \mathbb{E}_{t-1}[\theta_t(a)] \tag{190}$$

*is originally defined by Eq. (52).*

*Proof.* First, note that,

$$\mathbb{E}_t[W_{t+1}(a)] = 0, \text{ for all } t \geq 0. \tag{191}$$

Using the update Eq. (22), we have,

$$W_{t+1}(a) = \theta_{t+1}(a) - \mathbb{E}_t[\theta_{t+1}(a)] \tag{192}$$

$$= \theta_t(a) + \eta \cdot (I_t(a) - \pi_{\theta_t}(a)) \cdot R_t(a_t) - \left(\theta_t(a) + \eta \cdot \pi_{\theta_t}(a) \cdot \left(r(a) - \pi_{\theta_t}^\top r\right)\right) \tag{193}$$

$$= \eta \cdot (I_t(a) - \pi_{\theta_t}(a)) \cdot R_t(a_t) - \eta \cdot \pi_{\theta_t}(a) \cdot \left(r(a) - \pi_{\theta_t}^\top r\right). \tag{194}$$

According to Eq. (1), we have,

$$|W_{t+1}(a)| \le 3\,\eta \cdot R_{\max}. \tag{195}$$

The conditional variance of noise is,

$$\mathrm{Var}[W_{t+1}(a)|\mathcal{F}_t] := \mathbb{E}_t[(W_{t+1}(a))^2] \tag{196}$$

$$\le 2\,\eta^2 \cdot \mathbb{E}_t[(I_t(a) - \pi_{\theta_t}(a))^2 \cdot R_t(a_t)^2] + 2\,\eta^2 \cdot \pi_{\theta_t}(a)^2 \cdot \left(r(a) - \pi_{\theta_t}^\top r\right)^2, \tag{197}$$

where the inequality is by $(a + b)^2 \le 2a^2 + 2b^2$. Next, we have,

$$\mathbb{E}_t[(I_t(a) - \pi_{\theta_t}(a))^2 \cdot R_t(a_t)^2] = \pi_t(a) \cdot (1 - \pi_t(a))^2 \cdot r(a)^2 + \sum_{a' \ne a} \pi_{\theta_t}(a') \cdot \pi_t(a)^2 \cdot r(a')^2 \tag{198}$$

$$\le R_{\max}^2 \cdot \left(\pi_t(a) \cdot (1 - \pi_t(a))^2 + (1 - \pi_t(a)) \cdot \pi_t(a)^2\right) \tag{199}$$

$$= R_{\max}^2 \cdot \pi_{\theta_t}(a) \cdot (1 - \pi_{\theta_t}(a)), \tag{200}$$

and,

$$\left|r(a) - \pi_{\theta_t}^\top r\right| = \left|\sum_{a' \ne a} \pi_{\theta_t}(a') \cdot (r(a) - r(a'))\right| \tag{201}$$

$$\le \sum_{a' \ne a} \pi_{\theta_t}(a') \cdot |r(a) - r(a')| \tag{202}$$

$$\le 2\,R_{\max} \cdot \sum_{a' \ne a} \pi_{\theta_t}(a') \tag{203}$$

$$= 2\,R_{\max} \cdot (1 - \pi_{\theta_t}(a)). \tag{204}$$

Combining Eqs. (196), (198) and (201), we have,

$$\mathrm{Var}[W_{t+1}(a)|\mathcal{F}_t] \le 2\,\eta^2 \cdot R_{\max}^2 \cdot \pi_{\theta_t}(a) \cdot (1 - \pi_{\theta_t}(a)) + 8\,\eta^2 \cdot R_{\max}^2 \cdot \pi_{\theta_t}(a)^2 \cdot (1 - \pi_{\theta_t}(a))^2 \tag{205}$$

$$\le 10\,\eta^2 \cdot R_{\max}^2 \cdot \pi_{\theta_t}(a) \cdot (1 - \pi_{\theta_t}(a)). \tag{206}$$

Let $X_{t+1}(a) := \frac{W_{t+1}(a)}{6\,\eta \cdot R_{\max}}$. Then we have,

$$|X_{t+1}(a)| \le 1/2, \text{ and} \tag{207}$$

$$\mathrm{Var}[X_{t+1}(a)|\mathcal{F}_t] \le \frac{5}{18} \cdot \pi_{\theta_t}(a) \cdot (1 - \pi_{\theta_t}(a)). \tag{208}$$

According to Lemma 4, there exists an event $\mathcal{E}_1$ such that $\Pr(\mathcal{E}_1) \ge 1 - \delta$, and when $\mathcal{E}_1$ holds,

$$\forall t: \left|\sum_{s=1}^t X_{s+1}(a)\right| \le 6\sqrt{(V_t(a) + 4/3)\log\left(\frac{V_t(a) + 1}{\delta}\right)} + 2\,\log(1/\delta) + \frac{4}{3}\log 3, \tag{209}$$

which implies that,

$$\forall t: \left|\sum_{s=1}^t W_{s+1}(a)\right| \le 36\,\eta\,R_{\max}\sqrt{(V_t(a) + 4/3)\log\left(\frac{V_t(a) + 1}{\delta}\right)} + 12\,\eta\,R_{\max}\,\log(1/\delta) + 8\,\eta\,R_{\max}\log 3, \tag{210}$$

where $V_t(a) := \frac{5}{18} \cdot \sum_{s=1}^t \pi_{\theta_s}(a) \cdot (1 - \pi_{\theta_s}(a))$. $\qquad\square$

**Lemma 8** (Bounded progress). *For any action $a \in [K]$ with $N_\infty(a) = \infty$, if there exists $c > 0$ and $\tau < \infty$, such that, for all $t \geq \tau$,*

$$\sum_{s=1}^{t} |P_s(a)| \geq c \cdot V_t(a), \tag{211}$$

*where $V_t(a)$ is defined in Eq. (189), and if also*

$$\sum_{t=1}^{\infty} |P_t(a)| < \infty, \tag{212}$$

*then we have, almost surely,*

$$\sup_{t \geq 1} |\theta_t(a)| < \infty. \tag{213}$$

*Proof.* According to Eq. (55) and triangle inequality, we have,

$$|\theta_t(a)| \leq \left| \mathbb{E}[\theta_1(a)] \right| + \left| \sum_{s=1}^{t} W_s(a) \right| + \left| \sum_{s=1}^{t-1} P_s(a) \right|. \tag{214}$$

According to Lemma 7, there exists an event $\mathcal{E}_1$, such that $\Pr(\mathcal{E}_1) \geq 1 - \delta$, and when $\mathcal{E}_1$ holds, we have, for all $t \geq \tau + 1$,

$$\left| \sum_{s=1}^{t} W_s(a) \right| \leq 36 \, \eta \, R_{\max} \sqrt{(V_{t-1}(a) + 4/3) \cdot \log\left(\frac{V_{t-1}(a) + 1}{\delta}\right)} + 12 \, \eta \, R_{\max} \, \log(1/\delta) + 8 \, \eta \, R_{\max} \log 3. \tag{215}$$

According to Eq. (212), as $t \to \infty$,

$$\left| \sum_{s=1}^{t-1} P_s(a) \right| \leq \sum_{s=1}^{t-1} |P_s(a)| < \infty. \tag{216}$$

According to Eqs. (211), (215) and (216), we have,

$$\left| \sum_{s=1}^{t} W_s(a) \right| < \infty. \tag{217}$$

Combining Eqs. (214), (216) and (217), we have, as $t \to \infty$,

$$|\theta_t(a)| < \infty. \tag{218}$$

Take any $\omega \in \mathcal{E} := \{N_\infty(a) = \infty\}$. Because $\mathbb{P}(\mathcal{E} \setminus (\mathcal{E} \cap \mathcal{E}_1)) \leq \mathbb{P}(\Omega \setminus \mathcal{E}_1) \leq \delta \to 0$ as $\delta \to 0$, we have that $\mathbb{P}$-almost surely for all $\omega \in \mathcal{E}$ there exists $\delta > 0$ such that $\omega \in \mathcal{E} \cap \mathcal{E}_1$ while Eq. (215) also holds for this $\delta$. Take such a $\delta$. We have, almost surely,

$$\sup_{t \geq 1} |\theta_t(a)| < \infty. \qquad \square$$

**Lemma 9** (Unbounded positive progress). *For any action $a \in [K]$ with $N_\infty(a) = \infty$, if there exists $c > 0$ and $\tau < \infty$, such that, for all $t \geq \tau$,*

$$P_t(a) > 0, \text{ and} \tag{219}$$

$$\sum_{s=\tau}^{t} P_s(a) \geq c \cdot V_t(a), \tag{220}$$

*where $V_t(a)$ is defined in Eq. (189), and if also,*

$$\sum_{t=\tau}^{\infty} P_t(a) = \infty, \tag{221}$$

*then we have, almost surely,*

$$\theta_t(a) \to \infty, \text{ as } t \to \infty. \tag{222}$$

*Proof.* According to Eq. (55),

$$\theta_t(a) = \mathbb{E}[\theta_1(a)] + \sum_{s=1}^{t} W_s(a) + \sum_{s=1}^{t-1} P_s(a). \tag{223}$$

According to Lemma 7, there exists an event $\mathcal{E}_1$, such that $\Pr(\mathcal{E}_1) \geq 1 - \delta$, and when $\mathcal{E}_1$ holds, we have, for all $t \geq \tau + 1$,

$$\sum_{s=1}^{t} W_s(a) \geq -36\, \eta\, R_{\max} \underbrace{\sqrt{(V_{t-1}(a) + 4/3) \cdot \log\left(\frac{V_{t-1}(a) + 1}{\delta}\right)}}_{\heartsuit} - 12\, \eta\, R_{\max}\, \log(1/\delta) - 8\, \eta\, R_{\max} \log 3.$$

$$\tag{224}$$

By Eq. (221), as $t \to \infty$,

$$\sum_{s=1}^{t-1} P_s(a) \to \infty, \tag{225}$$

and the speed of $\sum_{s=1}^{t-1} P_s(a) \to \infty$ is strictly faster than $\heartsuit \to \infty$, according to Eq. (220). This implies that $\theta_t(a) \to \infty$, as a result of "cumulative progress" dominates "cumulative noise".

Take any $\omega \in \mathcal{E} := \{N_\infty(a) = \infty\}$. Because $\mathbb{P}(\mathcal{E} \setminus (\mathcal{E} \cap \mathcal{E}_1)) \leq \mathbb{P}(\Omega \setminus \mathcal{E}_1) \leq \delta \to 0$ as $\delta \to 0$, we have that $\mathbb{P}$-almost surely for all $\omega \in \mathcal{E}$ there exists $\delta > 0$ such that $\omega \in \mathcal{E} \cap \mathcal{E}_1$ while Eq. (224) also holds for this $\delta$. Take such a $\delta$. We have, almost surely,

$$\theta_t(a) \to \infty, \text{ as } t \to \infty. \qquad \square$$

**Lemma 10** (Unbounded negative progress). *For any action $a \in [K]$ with $N_\infty(a) = \infty$, if there exists $c > 0$ and $\tau < \infty$, such that, for all $t \geq \tau$,*

$$P_t(a) < 0, \text{ and} \tag{226}$$

$$-\sum_{s=\tau}^{t} P_s(a) \geq c \cdot V_t(a), \tag{227}$$

*where $V_t(a)$ is defined in Eq. (189), and if also,*

$$\sum_{t=\tau}^{\infty} P_t(a) = -\infty, \tag{228}$$

*then we have, almost surely,*

$$\theta_t(a) \to -\infty, \text{ as } t \to \infty. \tag{229}$$

*Proof.* The proof follows almost the same arguments for Lemma 9. $\qquad \square$

**Lemma 11** (Positive progress). *For any action $a \in [K]$ with $N_\infty(a) = \infty$, if there exists $c > 0$ and $\tau < \infty$, such that, for all $t \geq \tau$,*

$$P_t(a) > 0, \text{ and} \tag{230}$$

$$\sum_{s=\tau}^{t} P_s(a) \geq c \cdot V_t(a), \tag{231}$$

*where $V_t(a)$ is defined in Eq. (189), then we have, almost surely,*

$$\inf_{t \geq 1} \theta_t(a) > -\infty. \tag{232}$$

*Proof.* **First case:** if $\sum_{t=1}^{\infty} P_t(a) < \infty$, then according to Lemma 8, we have, almost surely,

$$\sup_{t \geq 1} |\theta_t(a)| < \infty, \tag{233}$$

which implies Eq. (232).

**Second case:** if $\sum_{t=1}^{\infty} P_t(a) = \infty$, then according to Lemma 9, we have, almost surely,

$$\theta_t(a) \to \infty, \text{ as } t \to \infty, \tag{234}$$

which also implies Eq. (232). $\qquad \square$

**Lemma 12** (Negative progress). *For any action $a \in [K]$ with $N_\infty(a) = \infty$, if there exists $c > 0$ and $\tau < \infty$, such that, for all $t \geq \tau$,*

$$P_t(a) < 0, \text{ and} \tag{235}$$

$$-\sum_{s=\tau}^{t} P_s(a) \geq c \cdot V_t(a), \tag{236}$$

*where $V_t(a)$ is defined in Eq. (189), then we have, almost surely,*

$$\sup_{t \geq 1} \theta_t(a) < \infty. \tag{237}$$

*Proof.* **First case:** if $-\sum_{t=1}^{\infty} P_t(a) < \infty$, then according to Lemma 8, we have, almost surely,

$$\sup_{t \geq 1} |\theta_t(a)| < \infty, \tag{238}$$

which implies Eq. (237).

**Second case:** if $-\sum_{t=1}^{\infty} P_t(a) = \infty$, then according to Lemma 10, we have, almost surely,

$$\theta_t(a) \to -\infty, \text{ as } t \to \infty, \tag{239}$$

which also implies Eq. (237). □

# C Rate of Convergence

**Theorem 3.** For a large enough $\tau > 0$, for all $T > \tau$, the average sub-optimality decreases at an $O\left(\frac{\ln(T)}{T}\right)$ rate. Formally, if $a^*$ is the optimal arm, then, for a constant $c$

$$\frac{\sum_{s=\tau}^{T} r(a^*) - \langle \pi_s, r \rangle}{T} \leq \frac{c \ln(T)}{T - \tau}$$

*Proof.* The progress for the optimal action is,

$$P_t(a^*) = \eta \cdot \pi_{\theta_t}(a^*) \cdot (r(a^*) - \pi_{\theta_t}^\top r) \tag{240}$$

$$\geq \eta \cdot \Delta \cdot \pi_{\theta_t}(a^*) \cdot \left(1 - \pi_{\theta_t}(a^*)\right). \qquad \left(\Delta := r(a^*) - \max_{a \neq a^*} r(a)\right) \tag{241}$$

$$\geq 0. \tag{242}$$

Since $\lim_{t \to \infty} \pi_{\theta_t}(a^*) = 1$, we have for all large enough $t \geq 1$,

$$\pi_{\theta_t}(a^*) \geq 1/2. \tag{243}$$

Since $N_\infty(a^*) = \infty$ (Theorem 2 ) and $|\mathcal{A}_\infty| \geq 2$ (Lemma 2), we have,

$$\sum_{t=1}^{\infty} (1 - \pi_t(a^*)) \geq \sum_{t=1}^{\infty} \pi_t(i_1) = \infty, \tag{244}$$

where $N_\infty(i_1) = \infty$ and $r(i_1) < r(a^*)$. Therefore, we have,

$$\sum_{t=1}^{\infty} P_t(a^*) = \infty. \tag{245}$$

The variance of noise is,

$$V_t(a^*) := \frac{5}{18} \cdot \sum_{s=1}^{t-1} \pi_{\theta_s}(a^*) \cdot (1 - \pi_{\theta_s}(a^*)), \tag{246}$$

which will be dominated by sum of $P_t(a^*)$ since $V_t(a^*)$ appears under square root (Lemma 7). Therefore, for all large enough $t \geq \tau$,

$$\theta_t(a^*) \geq C \cdot \sum_{s=\tau}^{t} (1 - \pi_s(a^*)). \tag{247}$$

On the other hand, we argued recursively (in the proofs for Theorem 2) that for all sub-optimal action $a \in [K]$ with $r(a) < r(a^*)$,

$$\sup_{t \geq 1} \theta_t(a) < \infty. \tag{248}$$

Therefore, we have, for all large enough $t \geq \tau$,

$$\theta_t(a^*) - \theta_t(a) \geq C \cdot \sum_{s=\tau}^{t} (1 - \pi_s(a^*)), \tag{249}$$

which implies that,

$$\sum_{a \neq a^*} \exp\{\theta_t(a) - \theta_t(a^*)\} \leq (K - 1) \cdot \exp\left\{-C \cdot \sum_{s=\tau}^{t} (1 - \pi_s(a^*))\right\}. \tag{250}$$

Therefore, we have,

$$1 - \pi_t(a^*) \leq \frac{1 - \pi_t(a^*)}{\pi_t(a^*)} = \sum_{a \neq a^*} \frac{\pi_t(a)}{\pi_t(a^*)} \leq (K - 1) \cdot \exp\left\{-C \cdot \sum_{s=\tau}^{t-1} (1 - \pi_s(a^*))\right\}. \tag{251}$$

Using Lemma 15 with $x_n = \sum_{s=\tau}^{t-1}(1 - \pi_s(a^*)) > 0$, $x_{n+1} = \sum_{s=\tau}^{t}(1 - \pi_s(a^*)) > 0$, $c = C > 0$ and $B = K - 1 \geq 1$ gives us that for all $t > \tau$,

$$\sum_{s=\tau}^{t}(1 - \pi_s(a^*)) \leq \frac{1}{C}\ln(Ct + e^{CM}) + \frac{\pi^2}{12C}, \tag{252}$$

where $M = \max\{K - 1, \frac{1}{C}\ln((K-1)C)), (1 - \pi_\tau(a^*))\} = K - 1$.

Finally, we use Eq. (252) to bound the average sub-optimality. For any $s \geq \tau$ and $T > \tau$,

$$r(a^*) - \langle \pi_s, r \rangle = \sum_{a \neq a^*} \pi_s\left[r(a^*) - r(a)\right] \leq 2\,R_{\max}\,(1 - \pi_s(a^*)).$$

$$(\text{Since } r(a) \in [-R_{\max}, R_{\max}])$$

Summing from $s = \tau$ to $T$,

$$\implies \frac{\sum_{s=\tau}^{T} r(a^*) - \langle \pi_s, r\rangle}{T} \leq \frac{2\,R_{\max}\left[\frac{1}{C}\ln(C\,T + e^{CM}) + \frac{\pi^2}{12C}\right]}{T - \tau}. \qquad \square$$

**Lemma 13.** *Let $\{y_n\}$ be the solution to the difference equation $y_{n+1} = y_n + B\,e^{-cy_n}$ with $B \geq 1$, $c > 0$ and $y_0 \geq \max(B, \frac{1}{c}\ln(B\,c))$. Let $\{x_n\}$ be a nonnegative valued sequence such that $x_0 \leq y_0$ and $x_{n+1} \leq x_n + B\,e^{-cx_n}$ for all $n \geq 0$. Then, $x_n \leq y_n$ for all $n \geq 0$.*

*Proof.* Define the function $f(y) = \max\{x + B\,e^{-cx} : 0 \leq x \leq y\}$. Clearly, $f$ is an increasing function of its argument.

Claim: For $y \geq \max(B, \frac{1}{c}\ln(B\,c))$, $f(y) = y + B\,e^{-cy}$. To prove this claim, for $x \in \mathbb{R}$, define

$$g(x) := x + B\,e^{-cx}. \tag{253}$$

Function $g$ is increasing on $(\frac{1}{c}\ln(B\,c), \infty)$ and decreasing on $(-\infty, \frac{1}{c}\ln(B\,c))$.

If $c < 1/B$, $\frac{1}{c}\ln(Bc) < 0$, hence $g$ is increasing on $(0, \infty)$. Hence, $f(y) = g(y)$, proving the claim.

If $c \geq 1/B$, then $\frac{1}{c}\ln(B\,c) \geq 0$. Hence, $g$ is decreasing on $(0, 1/c\ln(B\,c))$ and then increasing on $(1/c\ln(B\,c))$. Since $y \geq \frac{1}{c}\ln(B\,c)$, $f(y) = \max(g(0), g(y))$. Since we also have $y \geq B > \frac{1}{c}\ln(B\,c)$, $g(y) \geq g(B) > B = g(0)$ and thus $f(y) = g(y)$, finishing the proof of the claim.

The difference equation for $y_n$ is $y_{n+1} = y_n + B\,e^{-cy_n}$. Since $y_0 \geq \max(B, \frac{1}{c}\ln(B\,c))$ and $y_n$ is increasing, we have $y_n \geq \max(B, \frac{1}{c}\ln(B\,c))$ for all $n$. Therefore, we can apply the equality from the previous claim to get

$$y_{n+1} = y_n + B\,e^{-cy_n} = f(y_n).$$

The sequence $(x_n)$ satisfies the inequality $x_{n+1} \leq x_n + B\,e^{-cx_n} \leq f(x_n)$.

Now, let us prove $x_n \leq y_n$ using induction. The base case is $x_0 \leq y_0$ (given). Assume $x_k \leq y_k$ for some $k \geq 0$. Since $f$ is increasing, $x_k \leq y_k$ implies $f(x_k) \leq f(y_k)$. Using the properties $x_{k+1} \leq f(x_k)$ and $f(y_k) = y_{k+1}$, we get $x_{k+1} \leq f(x_k) \leq f(y_k) = y_{k+1}$. By the principle of mathematical induction, $x_n \leq y_n$ for all $n \geq 0$. $\qquad \square$

**Lemma 14.** *Let $c > 0$ and $B \geq 1$, and let $\{y_n\}_{n=0}^{\infty}$ be a sequence defined by the recurrence relation*

$$y_{n+1} = y_n + B\,e^{-cy_n}$$

*s.t. $y_0 \geq \max(B, \frac{1}{c}\ln(B\,c))$. Then,*

$$y_n \leq \frac{1}{c}\ln(cn + e^{cy_0}) + \frac{\pi^2}{12c}.$$

*Proof.* Define the function:

$$y(t) := \frac{1}{c}\ln(ct + e^{cy_0}), \qquad t \geq 0,$$

and note that it is an increasing function, Moreover, $\dot{y}(t)(= \frac{d}{dt}y(t)) = e^{-cy(t)}$ for any $t \geq 0$. Hence,

$$y(t) = y_0 + \int_0^t e^{-cy(s)}\,ds\,, \qquad t \geq 0$$

Define the function:

$$g(x) := x + B\,e^{-cx}\,,$$

and note that $g$ is increasing when $y \geq \frac{1}{c}\ln(B\,c)$. Moreover, $y(0) = y_0$ and $y_{n+1} = g(y_n)$.

Since $y(0) = y_0 \geq \frac{1}{c}\ln(B\,c)$ and both $\{y_n\}$ and $y(t)$ are increasing, $y_n \geq \frac{1}{c}\ln(B\,c)$ for all $n$ and , $y(t) \geq \frac{1}{c}\ln(B\,c)$ for all $t \geq 0$.

We first prove that

$$y(n) \leq y_n\,, \qquad n = 0, 1, \ldots\,. \tag{254}$$

We prove Eq. (254) by induction. The claim holds for $n = 0$ by construction. Now assume that $y(n) \leq y_n$ holds for some $n \geq 0$. Let us show that that $y(n+1) \leq y_{n+1}$ also holds. For this note that

$$\begin{aligned}
y(n+1) &= y(n) + \int_n^{n+1} e^{-cy(s)}ds \\
&\leq y(n) + \int_n^{n+1} e^{-cy(n)}ds && (t \mapsto y(t) \text{ is increasing}) \\
&= y(n) + e^{-cy(n)} \\
&\leq y(n) + B\,e^{-cy(n)} && (\text{since } B \geq 1) \\
&= g(y(n)) && (\text{definition of } g) \\
&\leq g(y_n) \\
&\quad (\text{induction hypothesis, } g \text{ is increasing for } y \geq \tfrac{1}{c}\ln(B\,c) \text{ and } y_n, y(n) \geq \tfrac{1}{c}\ln(B\,c)) \\
&= y_{n+1}\,. && (\text{By definition of } y_{n+1})
\end{aligned}$$

By the principle of mathematical induction, $y(n) \leq y_n$ for all $n \geq 0$.

Define $\Delta_n := y_n - y(n)$. From our previous inequality, we know that $\Delta_n \geq 0$. We now show that $\{\Delta_n\}_n$ is bounded, from which the desired statement follows immediately. To show that $\{\Delta_n\}_n$ is bounded we will show that $\Delta_{n+1} - \Delta_n$ is summable. To show this, we start by obtaining an expression for $\Delta_{n+1} - \Delta_n$. Let $A = e^{cy_0}$. Let $n \geq 0$. Direct calculation gives

$$\Delta_{n+1} - \Delta_n = e^{-cy_n} - \frac{1}{c}\ln\left(1 + \frac{c}{cn + A}\right)\,.$$

Using that for all $x > 0$, $\ln(1+x) \geq \frac{x}{1 + \frac{x}{2}}$, we get

$$\begin{aligned}
\Delta_{n+1} - \Delta_n &\leq e^{-cy_n} - \frac{1}{cn + A + \frac{c}{2}} \\
&\leq e^{-cy(n)} - \frac{1}{cn + A + \frac{c}{2}} && (\text{from Eq. (254)}) \\
&\leq \frac{1}{cn + A} - \frac{1}{cn + A + \frac{c}{2}} && (\text{definition of } y(n)) \\
&= \frac{1}{2c}\frac{1}{(n + A/c)(n + A/c + \frac{1}{2})} \\
&\leq \frac{1}{2c}\frac{1}{(n+1)^2}\,. && (\text{Since } y_0 \geq \tfrac{1}{c}\ln(B\,c)), A/c = B \geq 1)
\end{aligned}$$

For a fixed $m > 0$, summing up the above inequality from $n = 0$ to $m - 1$,

$$\Delta_m - \Delta_0 \leq \frac{1}{2c}\sum_{n=0}^{m-1}\frac{1}{(n+1)^2} \leq \frac{\pi^2}{12c} \qquad (\text{Since } \sum_{i=1}^{\infty}\tfrac{1}{i^2} = \tfrac{\pi^2}{6})$$

$$\implies \Delta_m \leq \frac{\pi^2}{12c}. \qquad (\text{Since } \Delta_0 = 0)$$

where $\pi = 3.14159\dots$. Hence, it follows that for any $n \geq 0$,

$$y_n = y(n) + \Delta_n \leq y(n) + \frac{\pi^2}{12c} = \frac{1}{c} \ln(cn + e^{cy_0}) + \frac{\pi^2}{12c}. \qquad \square$$

**Lemma 15.** *Let $\{x_n\}$ be a nonnegative valued sequence such that $x_{n+1} \leq x_n + B e^{-cx_n}$ for all $n \geq 0$ with $B \geq 1$, $c > 0$. Then, for all $n \geq 0$,*

$$x_n \leq \frac{1}{c} \ln(cn + e^{cM}) + \frac{\pi^2}{12c},$$

*where $M = \max\{B, \frac{1}{c} \ln(B c)), x_0\}$.*

*Proof.* Let $\{y_n\}$ be the solution to the difference equation $y_{n+1} = y_n + B e^{-cy_n}$ where $y_0 = \max\{B, \frac{1}{c} \ln(B c)), x_0\}$. Since $y_0 \geq x_0$ and $y_0 \geq \max(B, \frac{1}{c} \ln(B c))$, we can use Lemma 13 to conclude that for all $n \geq 0$,

$$x_n \leq y_n.$$

Furthermore, using Lemma 14 we can conclude that,

$$y_n \leq \frac{1}{c} \ln(cn + e^{cy_0}) + \frac{\pi^2}{12c}.$$

Combining the above inequalities completes the proof. $\qquad \square$

# D  Additional simulation results

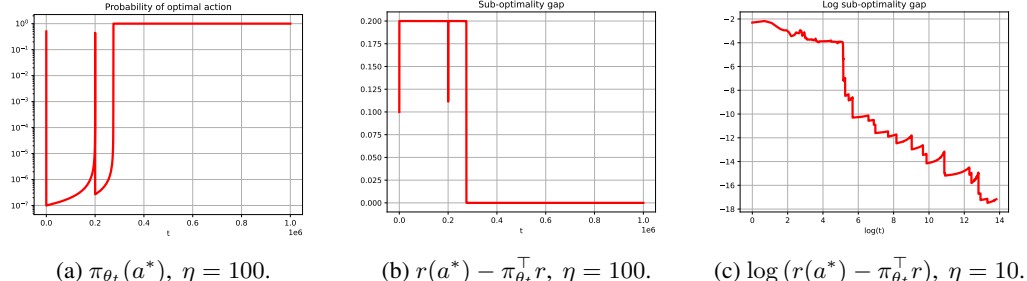

(a) $\pi_{\theta_t}(a^*)$, $\eta = 100$.   (b) $r(a^*) - \pi_{\theta_t}^\top r$, $\eta = 100$.   (c) $\log\left(r(a^*) - \pi_{\theta_t}^\top r\right)$, $\eta = 10$.

Figure 2: Visualization in a two-action stochastic bandit problem. Here the rewards are defined as $(-0.05, -0.25)$. Other details are same as for Figure 1. Figures 2a and 2a are based on a single run, while Figure 2c averages across 10 runs. Note that $\log\left(r(a^*) - \pi_{\theta_t}^\top r\right) \approx 10^{-33}$ at the final stages on Figure 2b.

