# OpenReview forum: "Small steps no more: Global convergence of stochastic gradient bandits for arbitrary learning rates"
_NeurIPS.cc/2024/Conference — NeurIPS 2024 poster_

### Official Review · Reviewer_4TYZ · 2024-07-12

**Soundness:** 3
**Presentation:** 3
**Contribution:** 3
**Rating:** 5
**Confidence:** 4

**Summary:**

This paper studies stochastic gradient bandits with arbitrarily large learning rates. The authors prove an interesting result -- the learning rate of the gradient bandit algorithm (in particular, REINFORCE without baseline) can be arbitrarily large. Some numerical simulations are also provided to validate the findings.

**Strengths:**

1. Theoretically, the authors prove that for any constant learning rate, the stochastic gradient bandit algorithm (REINFORCE without baselines) converges to the globally optimal policy almost surely as the number of iterations goes to infinity. The authors also avoid using non-uniform smoothness and noise growth conditions in a previous related work ([1]).

2. The authors also provide some simulation results to verify their theoretical findings.



[1] Stochastic gradient succeeds for bandits.

**Weaknesses:**

1. Optimization with large learrning rates have been studied, but the authors seem to only focus on the RL settings. It might be helpful to also include some literature review on deep learning with large learning rates (see, e.g., [1]).

2. Although it is shown that the algorithm can converge with arbitrarily large learning rates, it is still unclear what the exact convergence rate is. In contrast, a previous work [2], as also mentioned by the authors, has the finite-time convergence guarantees.

3. The code is not provided, although the experimental setting is simple.

[1] Gradient Descent on Neural Networks Typically Occurs at the Edge of Stability

[2] Stochastic gradient succeeds for bandits.

**Questions:**

1. Algorithm 1 is the gradient bandit algorithm without baselines. Can we also have similar results for the version with baselines?

2. What would be the challenges of extending the current analysis to the non-asymptotic rate analysis? Furthermore, what would be the challenges of extending the current proof techniques to other RL algorithms like natural gradient methods?

**Limitations:**

1. The convergence results only imply asymptotic convergence without an explicit rate. Thus one direct question one may ask is, how does the learning rate affects the convergence speed. It would be interesting to study if there is an optimal learning rate.

2. The current results are only limited to REINFORCE without baselines for bandit setting, which is simpler than more complex RL settings. Hence it is unclear if it can provide any guidance to RL research in general.

---

> ### Author Rebuttal · Authors · 2024-08-07
>
> We appreciate that the reviewer recognized the contribution of the work. We answer the questions as follows.
>
> >**include some literature review on deep learning with large learning rates**
>
> Thank you for pointing the "edge of stability" paper to us. We will cite this line of work in the related work discussion starting from Line 104 in the paper.
>
> >**what the exact convergence rate is**
>
> Please find a detailed discussion in the common rebuttal, regarding why the techniques in [2] are not applicable and cannot be used here to obtain a rate.
>
> >**The code is not provided**
>
> We will upload a link for running the simulations in the updated version of the paper.
>
> >**Can we also have similar results for the version with baselines?**
>
> Thank you for asking this interesting question.
>
> **First**, our preliminary calculations show that similar results can be obtained for the version with action-independent baselines, such as $\pi_{\theta_t}^\top r$.
>
> **Second**, in Algorithm 1, if we replace $R_t(a_t)$ with $R_t(a_t) - \pi_{\theta_t}^\top r$ (or $R_t(a_t) - B_t$ with any other action-independent baseline $B_t \in \mathbb{R}$), then Lemmas 1 and 2 still hold because of $R_t(a_t) - \pi_{\theta_t}^\top r$ is still bounded (by Eq. (1)). Theorem 1 will hold because of the progress is exactly the same as in Eqs. (11) and (13), according to the well known result that action-independent baselines contribute $0$ to policy gradient, meaning that Proposition 1 holds for any action-independent baselines.
>
> >**What would be the challenges of extending the current analysis to the non-asymptotic rate analysis?**
>
> **First**, the challenge is that monotonic improvement (in expectation) over $\pi_{\theta_t}^\top r$ cannot be shown for large learning rates (non-monotonicity of $\pi_{\theta_t}^\top$ is observed in simulations in Fig. 1 in the main paper, also as pointed in Line 304), unlike Lemma 4.6 of [23] (when learning rate is very small). Therefore, it is not clear how to quantify the progress (how much $r(a^*) - \pi_{\theta_t}^\top r$ is reduced in expectation after one stochastic gradient step). Without quantifying the (expected) progress, it seems difficult to do non-asymptotic rate analysis.
>
> **Second**, there exist analyses for non-monotonic improvement over objective functions (such as Nesterov's accelerated gradient), but for convex functions. However, here we have a non-concave maximization (Line 95).
>
> Please also find a detailed discussion in the common rebuttal.
>
> >**what would be the challenges of extending the current proof techniques to other RL algorithms like natural gradient methods?**
>
> Natural gradient methods achieve faster convergence results than standard policy gradient when using exact gradient updates. However, with stochastic on-policy sampling $a_t \sim \pi_{\theta_t}(\cdot)$, if we use constant learning rates $\eta \in \Theta(1)$ for natural policy gradient method, then it can fail, by converging to sub-optimal deterministic policies with  positive probability (Theorem 3 of [19] and Proposition 1 of Chung2021).
>
> **[Chung2021]** Beyond variance reduction: Understanding the true impact of baselines on policy optimization, in ICML 2021.

---

> > ### Comment · Reviewer_4TYZ · 2024-08-11
> > **Reply to the rebuttal**
> >
> > Thank you for your rebuttal. Your rebuttal addressed my questions and I would like to keep my score to vote for acceptance.

---

### Official Review · Reviewer_qHxg · 2024-07-12

**Soundness:** 3
**Presentation:** 3
**Contribution:** 3
**Rating:** 6
**Confidence:** 4

**Summary:**

This work studies the asymptotic global convergence rate of the stochastic gradient bandit algorithm with an arbitrary constant learning rate and proves that this algorithm asymptotically converges to the global optimal. This work reveals how this algorithm balances exploitation and exploration and proves the results by contradiction and reduction. This work also provides simulation experiments to support their results.

**Strengths:**

The proof process based on contradiction and reduction is novel. Furthermore, the analysis reveals why the stochastic gradient bandits algorithm can naturally balance exploitation and exploration, which deepens the understanding of this algorithm.

**Weaknesses:**

The empirical success of a large learning rate is unclear. This work shows that softmax policies and logistic regression can be learned when using a large learning rate. However, they do not show that a large learning rate can lead to great empirical performance.

**Questions:**

Question 1: As shown in Figure 1, when choosing $\eta= 100$ or $1000$, the algorithm does not converge. Can you discuss this phenomenon in detail?

Question 2: [1] consider $K=10$ in the simulation study. However, this work does simulation experiments with small arm numbers such as $K=4$ or $K=2$. It would be better to consider larger $K$ and show that the learning rate would not be influenced by $K$.

Comment 1: It would be helpful to discuss the challenge when using the technique in [1] for a constant learning rate.

[1] Mei, J., Zhong, Z., Dai, B., Agarwal, A., Szepesvari, C., & Schuurmans, D. (2023, July). Stochastic gradient succeeds for bandits. In International Conference on Machine Learning (pp. 24325-24360). PMLR.

**Limitations:**

The authors adequately addressed the limitations and societal impact.

---

> ### Author Rebuttal · Authors · 2024-08-07
>
> We appreciate that the reviewer understood and recognized the contribution of the work. We answer the questions as follows.
>
> >**$\eta = 100$ and $\eta = 1000$ do not converge. Can you discuss this phenomenon in detail?**
>
> We ran more iterations for $\eta = 100$ and $\eta = 1000$, and eventually all runs converged. Please see Fig. 1 in the rebuttal pdf for results. A new observation is that those curves converge when the optimal action is sampled, i.e., $a_t = a^*$. This is aligned with theory, since the first part of Theorem 2 (Line 241) proved that the optimal action will be sampled for infinitely many times as $t \to \infty$.
>
> >**larger $K = 10$**
>
> We ran experiments using $K = 10$ as suggested for learning rates $\eta \in \\{1, 10\\}$, extending the example in the main paper to $r = (0.2, 0.05, -0.1, -0.4, -0.5, -0.6, -0.7, -0.8, -0.9, -1.0)^\top$. Please see Fig. 2 in the rebuttal pdf for results.
>
> >**It would be helpful to discuss the challenge when using the technique in [1] for a constant learning rate.**
>
> **First**, monotonic improvement over $\pi_{\theta_t}^\top r$ cannot be established for large learning rates. In fact, for large learning rates, non-monotonicity of $\pi_{\theta_t}^\top$ is observed in Fig. 1 (metioned in Line 304).
>
> **Second**, in particular, Lemma 4.6 in [1] requires very small constant learning rates ($\eta \le \frac{\Delta^2}{40 \cdot K^{3/2} \cdot R_{\max}^3 }$), which follows from their Lemma 4.2 (smoothness) and Lemma 4.3 (noise growth condition). Using large learning rates makes those two lemmas not applicable, and monotonic improvement of $\pi_{\theta_t}^\top r$ is no longer assured.

---

> > ### Comment · Reviewer_qHxg · 2024-08-12
> >
> > Thank you for your rebuttal. The additional experiments and the discussion on the technique challenge address my concerns. I will keep my positive score to support this valuable work.

---

### Official Review · Reviewer_E4oN · 2024-07-13

**Soundness:** 3
**Presentation:** 3
**Contribution:** 3
**Rating:** 8
**Confidence:** 3

**Summary:**

The paper presents a novel theoretical analysis of the stochastic gradient bandit algorithm, showing that it converges to a globally optimal policy almost surely using any constant learning rate. This result is significant as it extends the understanding of stochastic gradient methods in bandit settings, even when standard smoothness and noise control assumptions do not hold.

**Strengths:**

1. Theoretical Contribution: The paper provides a strong theoretical result, proving the global convergence of the stochastic gradient bandit algorithm for any constant learning rate. This is a significant advancement over existing literature that typically requires decaying or small learning rates.
2. Novel Insights: The authors uncover interesting properties of action sampling rates and the relationship between cumulative progress and noise, contributing to a deeper understanding of exploration-exploitation trade-offs in stochastic gradient methods.
3. Clarity and Rigor: The proofs are presented with clarity and rigor, making the paper accessible to readers with a solid background in stochastic optimization and reinforcement learning.
4. Empirical Validation: The simulation studies support the theoretical findings, demonstrating the convergence behavior under various learning rates.

**Weaknesses:**

1. Practical Implications: While the theoretical results are robust, the practical implications, especially regarding the choice of learning rate in real-world applications, are not thoroughly discussed. It would be beneficial to include more guidance on how practitioners can leverage these findings.
2. Rate of Convergence: The paper establishes almost sure convergence but does not provide specific rates of convergence. Including more detailed analysis or conjectures about the rate would enhance the practical utility of the results.
3. Generalization: The study is limited to multi-armed bandits. Extending the results to more general reinforcement learning settings would significantly increase the impact of the work.
4. Assumption 1: The assumption that the true mean reward has no ties (Assumption 1) is restrictive. Addressing this limitation or providing more discussion on how this assumption might be relaxed in future work would strengthen the paper.

**Questions:**

Proof Details:

Could you provide more detailed explanations for Lemma 1 and Lemma 2? Specifically, how do the assumptions ensure the boundedness of parameters?
Experimental Results:

In the experimental section, you use different learning rates (η). Could you elaborate on the observed differences in convergence behavior for these learning rates? How does the algorithm perform with learning rates outside the tested range?
Practical Implications:

How would you recommend practitioners choose an appropriate learning rate for real-world applications based on your findings? Are there any heuristics or rules of thumb that can be derived from your results?
Generalization:

Your results are currently limited to multi-armed bandits. What challenges do you foresee in extending these results to more general reinforcement learning settings? How might the approach change?
Assumption 1:

Assumption 1 states that the true mean reward has no ties. How critical is this assumption to your results? Do you have any ideas on how to relax this assumption in future work?
Convergence Rate:

While you have shown almost sure convergence, the specific rate of convergence is not detailed. Could you provide more insights or conjectures about the expected rate of convergence for different learning rates?
Limitations and Future Work:

You mention that a more refined analysis is needed to explain the subtleties of different stages of convergence. Could you suggest specific directions or methods for this refined analysis?

---

> ### Author Rebuttal · Authors · 2024-08-07
>
> We appreciate that the reviewer understood and recognized the contribution of the work, and we thank the reviewer for carefully reading and checking the results. The main concerns are addressed as follows.
>
> >**detailed explanations for Lemma 1 and Lemma 2**
>
> According to Eq. (1), the sampled reward $R_t(a_t)$ is in a bounded range of $[-R_{\max}, R_{\max}]$ with $R_{\max} < \infty$. By design, we use a constant learning rate $\eta < \infty$. This argument is shown in Eqs. (26) and (27) for Lemma 1 in the appendix.
>
> >**elaborate on the observed differences in convergence behavior for these learning rates**
>
> From the simulations, smaller $\eta$ values such as $1$ and $10$ always enter the final stage of $\pi_{\theta_t}(a^*)$ being close to $1$ more quickly, while larger $\eta$ values ($100$ and $1000$) can reduce the sub-optimality gap $r(a^*) - \pi_{\theta_t}^\top r$ to much smaller values when $\pi_{\theta_t}$ is already in the final stage of $\pi_{\theta_t}(a^*) \approx 1$. However, there is a trade off, such that larger $\eta$ values (on average) spend a longer time entering the final stage.
>
> >**learning rates outside the tested range**
>
> We ran experiments on the same example in the paper using $\eta = 0.01$ and $\eta = 0.1$, which outside the tested range of $\{1, 10, 100, 1000\}$ (since $1000$ is already a super large learning rate, we did not test $\eta > 1000$). Please see Fig. 3 in the rebuttal pdf for results.
>
> >**How would you recommend practitioners choose an appropriate learning rate...Are there any heuristics or rules of thumb...?**
>
> In deep learning, large learning rates have been applied empirically. For example, [Li2020] showed that increasing learning rates can achieve good performance for networks with BatchNorm, which is similar to what we speculated for RL (Line 345).
>
> However, multiple factors affect practical performance. Our heuristic is that when the optimal solution is deterministic (no entropy regularization), then stochastic policy gradient could work with large learning rates. If there is entropy regularization, then eventually the learning rate has to be decayed, since otherwise oscillation can happen (optimal solution is not deterministic, and overshooting can happen).
>
> **[Li2020]** An Exponential Learning Rate Schedule for Deep Learning, in ICLR 2020.
>
> >**What challenges do you foresee in extending these results to more general reinforcement learning settings? How might the approach change?**
>
> **First**, currently we do not know if the key properties we established for bandits (Lemmas 1 and 2) will hold in MDPs or not, which is the key challenge of extending to general reinforcement learning settings.
>
> **Second**, in general MDPs, exploration over the state space is a problem that does not exist in bandits [1], which requires the stochastic gradient methods to be modified accordingly.
>
> >**Assumption 1 ... How critical is this assumption ... how to relax this assumption in future work?**
>
> **First**, Assumption 1 is for the simplicity of our arguments. It shouldn't be critical, since one can collapse actions with the same rewards without changing the optimal expected reward / value.
>
> **Second**, regarding how to relax this assumption, please refer to the common rebuttal for a more detailed discussion.
>
> >**Could you suggest specific directions or methods for this refined analysis**
>
> We speculate that the established asymptotic convergence of $\pi_{\theta_t}(a^*) \to 1$ as $t \to \infty$ can be split into two phases: early and final stages, corresponding to when $\pi_{\theta_t}(a^*) < 1 - \epsilon$ or $\pi_{\theta_t}(a^*) \ge 1 - \epsilon$ respectively.
>
> The phase transition happens at the time point $t_0 < \infty$ such that for all $t \ge t_0$, we have $\pi_{\theta_t}(a^*) \ge 1 - \epsilon$.  Therefore, a refined analysis would be to characterize how the time $t_0$ depends on learning rates, as well as other problem specific quantities, such as reward gap and reward range.
>
> >**Could you provide more insights or conjectures about the expected rate of convergence for different learning rates?**
>
> **First**, for all learning rates, from Figs. 1(a) and 1(b) in the main paper,  $\log{ (r(a^*) - \pi_{\theta_t}^\top r) } \approx - \log{t} + c$ (from numerical values,  slopes are $\approx -1$), which means that $r(a^*) - \pi_{\theta_t}^\top r \approx C/t$, i.e., the rate of convergence is about $O(1/t)$ (as mentioned in Line 315).
>
> **Second**, for different learning rates, we speculate that the time spent in early stage scales with $\eta$, such as $O(\eta)$, while the rate in final stage scales with $1/\eta$, such as $O(1/(\eta \cdot t))$, aligned with previous observed differences in convergence behavior for different learning rates in Fig. 1 in the main paper.

---

### Official Review · Reviewer_jZkA · 2024-07-14

**Soundness:** 3
**Presentation:** 4
**Contribution:** 3
**Rating:** 5
**Confidence:** 4

**Summary:**

The paper reveals that the stochastic gradient bandit algorithm converges to a globally optimal policy almost surely using any constant learning rate. This result stands even when traditional smoothness and noise control assumptions are not met, showing the algorithm’s balance between exploration and exploitation.

**Strengths:**

1. The paper provides useful asymptotic insights into stochastic gradient bandits, which to the best of my knowledge are novel.  Unlike previous work, where the methodology would not work for large learning rates,  the gradient bandit algorithm proposed by the authors is proven to asymptotically converge to the globally optimal policy.
2. The paper discusses how exploration and exploitation trade-offs are balanced without exploration bonuses. Significant insights are provided into how constant learning rates affect the algorithm’s ability to explore different actions and avoid getting stuck in sub-optimal points.
3. The proof sketches for Theorems 1 and 2 are particularly beneficial in elucidating the intuition underpinning the proofs. The contradiction-based argument presented in the case where ( $K \geq 2$) is especially neat and clever.

**Weaknesses:**

1. As the authors also acknowledge, Assumption 1 seems to be a strong one and possibly unrealistic in applications.
2. The work establishes almost sure convergence to the globally optimal policy as $t \rightarrow \infty$, which doesn't tell anything about the rate of convergence.
3. Section 5.1 in [23] appears to have similar results as the one presented in this paper. In the discussion of the natural exploratory behavior is attributed to the softmax Jacobian in the update, which is also discussed extensively in [23]. However, I understand that the main focus of  [23] is to show that the stochastic gradient bandit algorithm converges to a globally optimal policy at an $O(1/t)$ rate. I am concerned that this paper loses significance as [23] already establishes convergence rates to the globally optimal policy.
4. Even though the main focus of the paper is theoretical, it would be nice to see how the stochastic gradient compares to other bandit algorithms such as UCB and Thompson Sampling in numerical experiments.


*[23] Mei, Jincheng, et al. "Stochastic gradient succeeds for bandits." International Conference on Machine Learning. PMLR, 2023.*

**Questions:**

1. The paper focuses on the simplest setting of the stochastic bandit problem, where decisions only matter for one step. The more common consideration in bandits literature is to maximizing the cumulative reward (or minimizing the expected regret). Can we apply the stochastic gradient bandit algorithm in that setting?
2. The numerical experiments section in the paper indicates that smaller learning rates perform better during early optimization stages, while larger rates are beneficial later. I have two related thoughts to this and was wondering if any comments could be made regarding the same:
- For adaptive learning rates, can one think of running the algorithm in batches, such that the learning rates are decreased systematically for consequent batches?
- Intuitively the right learning rate should depend on the underlying difficulty of the reward generating mechanism for different arms, however I am not sure if that is apparent from the discussion in the paper.
3. Can this be extended to the contextual bandits framework?

**Limitations:**

Yes, limitations have been adequately addressed. Societal impact statement N/A.

---

> ### Author Rebuttal · Authors · 2024-08-07
>
> We appreciate that the reviewer understood and recognized the contribution of the work. We answer the questions as follows.
>
> Please refer to the common rebuttal for questions regarding Assumption 1, rate of convergence, and comparison with [23].
>
> >**...maximizing the cumulative reward (or minimizing the expected regret). Can we apply the stochastic gradient bandit algorithm in that setting?**
>
> In fact, if we can obtain a rate of convergence (say $O(1/t)$ as speculated in the common rebuttal), such that $\mathbb{E}{ [ r(a^*) - \pi_{\theta_t}^\top r]} \le C/t$, then this will imply a expected regret upper bound by summing up sub-optimality gap,
> $$\sum_{t=1}^{T}{ \Big( r(a^*) - \mathbb{E}{[\pi_{\theta_t}^\top r]} \Big) } \le \sum_{t=1}^{T}{C/t} \le C \cdot ( \log{T} + 1 ),$$
> which means that to get such a result, it is sufficient to prove a rate of convergence result.
>
> >**adaptive learning rates...decreased systematically for consequent batches?**
>
> **First**, we ran experiments using adaptive decaying learning rates on the same example in the paper. The total number of iterations is $T = 2 \times 10^6$. For the first $10^6$ iterations, we use a linearly decayed learning rate from $\eta = 1000$ (or $\eta = 100$) to $\eta = 1$ at $t = 10^6$, i.e., $\eta_t = \mathbf{1000} \times (1 - t/10^6) + t/10^6$ (or replace $\mathbf{1000}$ with $\mathbf{100}$) for all $t \in [1, 10^6]$. For the second $10^6$ iterations, we decay the learning rate as $O(1/\sqrt{t})$, i.e., $\eta_t = 1/\sqrt{t-10^6}$ for all $t \in [10^6 + 1, 2 \times 10^6]$. Please see Fig. 4 in the rebuttal pdf for results.
>
> **Second**, we believe that this decaying learning rate scheme will lead to slower convergence speeds than using constant learning rates, as suggested by the analysis carried out in the paper of Lu2024.
>
> **[Lu2024]** Towards Principled, Practical Policy Gradient for Bandits and Tabular MDPs.
>
> >**Intuitively the right learning rate should depend on the underlying difficulty of the reward generating mechanism for different arms, however I am not sure if that is apparent from the discussion in the paper.**
>
> Thank you for pointing this out. We agree with the reviewer on their intuition and insights.
>
> **First**, we would re-emphasize that the asymptotic convergence result  is the very first result for using large learning rates in stochastic gradient bandits, which is already difficult to obtain.
>
> **Second**, asymptotic convergence result itself without any characterization of rates is not enough to tell the difference in dependence on problem difficulty. As mentioned in Line 335 in the paper, a more refined characterization of convergence behaviors is needed to reflect this dependence.
>
> >**Can this be extended to the contextual bandits framework?**
>
> Thank you for asking, and we are also pursuing this interesting direction. It is doable but not immediate. Extra work needs to be done for analyzing stochastic gradients using linear features (rather than the softmax tabular parameterizaion here), which is of independent of interest.

---

> > ### Comment · Reviewer_jZkA · 2024-08-12
> >
> > I appreciate the authors' detailed response to my questions. After reading the rebuttal, I decide to maintain my initial score.

---

### Official Review · Reviewer_RHGg · 2024-07-14

**Soundness:** 2
**Presentation:** 2
**Contribution:** 2
**Rating:** 5
**Confidence:** 2

**Summary:**

This paper proved that stochastic gradient bandits converge to a globally optimal policy almost surely for arbitrary constant learning rates if true mean reward has no ties.

**Strengths:**

This work extends the previous convergence results of stochastic gradient bandits by generalizing from a specific constant learning rate to an arbitrary constant learning rate. The high-level idea is demonstrated, and simple simulations are conducted to validate the theoretical finding.

**Weaknesses:**

This paper heavily relies on prior work [23], which established the asymptotic rate for a constant learning rate. As a follow-up study, however, I believe this paper does not present sufficiently strong results to warrant acceptance at a top conference.

**Questions:**

In simulations, if the learning rate is initially set to large number and then decreased, does algorithm show fast convergence?

**Limitations:**

Yes, the authors address the limitations.

---

> ### Author Rebuttal · Authors · 2024-08-07
>
> We thank the reviewer for taking time to review our work. We hope the following can help clarify matters.
>
> >**heavily relies on prior work [23]**
>
> This is simply incorrect. Our analysis is significantly different from [23], which is built on smoothness and growth condition, while our results do not exploit. Please refer to the common rebuttal for a detailed explanation.
>
> >**In simulations, if the learning rate is initially set to large number and then decreased, does algorithm show fast convergence?**
>
> **First**, we ran experiments using adaptive decaying learning rates as suggested on the same example in the paper. The total number of iterations is $T = 2 \times 10^6$. For the first $10^6$ iterations, we use a linearly decayed learning rate from $\eta = 1000$ (or $\eta = 100$) to $\eta = 1$ at $t = 10^6$, i.e., $\eta_t = \mathbf{1000} \times (1 - t/10^6) + t/10^6$ (or replace $\mathbf{1000}$ with $\mathbf{100}$) for all $t \in [1, 10^6]$. For the second $10^6$ iterations, we decay the learning rate as $O(1/\sqrt{t})$, i.e., $\eta_t = 1/\sqrt{t-10^6}$ for all $t \in [10^6 + 1, 2 \times 10^6]$. Please see Fig. 4 in the rebuttal pdf for results.
>
> **Second**, we believe that this decaying learning rate scheme will lead to slower convergence speeds than using constant learning rates, as suggested by the analysis carried out in the paper of Lu2024.
>
> **[Lu2024]** Towards Principled, Practical Policy Gradient for Bandits and Tabular MDPs.

---

> > ### Comment · Reviewer_RHGg · 2024-08-12
> > **Reply to rebuttal**
> >
> > Thank you for your rebuttal. I have updated the score to 'borderline accept.'

---

### Author Rebuttal · Authors · 2024-08-07

We thank the reviewers for their valuable comments and recognition of the contributions. This common feedback answers questions raised by multiple reviewers.

>**Comparison to [23] (Reviewers RHGg, jZkA, qHxg, 4TYZ)**

**First**, we would like to emphasize that the asymptotic convergence arguments in this paper go well beyond \emph{any existing results and techniques}. The level of difficulty is paramount, since the foundational conditions (smoothness and growth condition [23]) for most existing optimization convergence and rate analysis are not applicable (as mentioned Line 116 in the paper). We believe that achieving novel results by developing innovative new proof techniques and uncovering new properties of stochastic gradients is a substantial achievement for whole optimization community, far beyond RL.

**Second**, the analysis in this paper provides new implications for practice of policy gradient methods. Previous results, including [23], do not cover practical choices for learning rates, yet the results in this paper can. For the example in Fig. 1 in the main paper, Lemma 4.6 of [23] gives $\eta \le \frac{\Delta^2}{40 \cdot K^{3/2} \cdot R_{\max}^3 } = 0.00007$ (Line 294), which is much smaller than any learning rate in Fig. 1, which are in the realm of the analysis in this paper.

**Third**, the analysis in [23] relies on monotonic improvement results over $\pi_{\theta_t}^\top r$ (in expectation) that \textbf{do not hold} in our setting and therefore are \textbf{not applicable}. In particular, Lemma 4.6 in [23] requires one to use very small constant learning rates ($\eta \le \frac{\Delta^2}{40 \cdot K^{3/2} \cdot R_{\max}^3 }$). Therefore, we had to develop new insights, which led to new technical, innovative results (Lemmas 1 and 2) that are distinctly different from [23], and to our knowledge completely novel.

>**Rate of convergence (Reviewers jZkA, E4oN, 4TYZ)**

**First**, we would like to emphasize that even asymptotic convergence is a totally new result for using large learning rates with stochastic gradient, without exploiting smoothness and growth conditions. As explained, in this case there is no monotonic improvement (in expectation) over $\pi_{\theta_t}^\top r$ (unlike [23]), and whether convergence (or oscillation) will occur asymptotically is not obvious.

**Second**, we speculate that the convergence rate is $O(1/t)$. Figs. 1(a) and 1(b) in the main paper show that $\log{ (r(a^*) - \pi_{\theta_t}^\top r) } \approx - \log{t} + c$ (from numerical values, the slopes are $\approx -1$), which means that $r(a^*) - \pi_{\theta_t}^\top r \approx C/t$, i.e., the rate of convergence is about $O(1/t)$ (mentioned in Line 315).

**Third**, the key difficulty arises from the non-monotonicity of $\pi_{\theta_t}^\top r$ (observed in Fig. 1, mentioned in Line 304).
Since Lemma 4.6 of [23] requires a very small $\eta$, which does not apply here, we cannot quantify the progress (how much $r(a^*) - \pi_{\theta}^\top r$ is reduced in expectation after one stochastic gradient update) using their techniques, which makes it unclear how to use results of [23] to obtain a rate here.

>**Assumption 1 (Reviewers jZkA, E4oN)**

**First**, we believe Algorithm 1 simply works without Assumption 1. To see intuition/evidence, consider $r = (0.9, 0.9, 0.7, 0.6, 0.6, 0.5)^\top$, which does not satisfy Assumption 1. Consider softmax PG with exact gradients, which achieves $\pi_{\theta_t}^\top r \to 0.9$ as $t \to \infty$. Now the question is if $\pi_{\theta_t}$ approaches a strict one-hot policy or not. Suppose $\theta_t(1) > \theta_t(2)$, then we have,
$$\theta_{t+1}(1) - \theta_{t+1}(2) = \theta_t(1) - \theta_t(2) + \eta \cdot (\pi_{\theta_t}(1) - \pi_{\theta_t}(2)) \cdot (0.9 - \pi_{\theta_t}^\top r) > \theta_t(1) - \theta_t(2),$$
which implies that $\theta_{t}(1) - \theta_{t}(2)$ is monotonically increasing if initially $\theta_1(1) - \theta_1(2) > 0$. As a consequence, $\pi_{\theta_t}(1) \to 1$ and $\pi_{\theta_t}(2) \to 0$, even if the two actions have the same reward $0.9$. This means except for a zero measure initialization such that $\pi_{\theta_1}(1) = \pi_{\theta_1}(2)$, the policy $\pi_{\theta_t}$ eventually goes to a strictly optimal one-hot policy (mentioned in Remark 1).

**Second**, to extend the above calculations from exact gradient to the stochastic online setting, extra work needs to be done to show that with probability $1$, only one action dominates the others, rather than they achieve an asymptotic balance to have mixed (rather than strictly one-hot) convergent policies.

---

### Decision · Program_Chairs · 2024-09-25

**Decision:**

Accept (poster)

**Comment:**

The most common score for this work is Borderline accept, and none of those reviewers make a strong case for its acceptance. However, after reviewing the paper and the commentary of Reviewer E4oN, I believe this paper contributes some novel proof techniques that alleviate undue dependence on unrealistic assumptions present in prior art. Therefore, it clears the bar for acceptance.